# Dendritic integration in olfactory bulb granule cells upon simultaneous multispine activation: Low thresholds for nonlocal spiking activity

**Max Mueller**📷*, **Veronica Egger**📷*

Neurophysiology, Institute of Zoology, Universität Regensburg, Regensburg, Germany

* Max1.Mueller@ur.de (MM); Veronica.Egger@ur.de (VE)

**Data Availability Statement:** All excel sheets of cumulative data and the IGOR experiment analysis files and image stacks of the example data used to create the figures and all excel sheets containing

## Abstract

The inhibitory axonless olfactory bulb granule cells form reciprocal dendrodendritic synapses with mitral and tufted cells via large spines, mediating recurrent and lateral inhibition. As a case in point for dendritic transmitter release, rat granule cell dendrites are highly excitable, featuring local $Na^+$ spine spikes and global $Ca^{2+}$- and $Na^+$-spikes. To investigate the transition from local to global signaling, we performed holographic, simultaneous 2-photon uncaging of glutamate at up to 12 granule cell spines, along with whole-cell recording and dendritic 2-photon $Ca^{2+}$ imaging in acute juvenile rat brain slices. Coactivation of less than 10 reciprocal spines was sufficient to generate diverse regenerative signals that included regional dendritic $Ca^{2+}$-spikes and dendritic $Na^+$-spikes (D-spikes). Global $Na^+$-spikes could be triggered in one third of granule cells. Individual spines and dendritic segments sensed the respective signal transitions as increments in $Ca^{2+}$ entry. Dendritic integration as monitored by the somatic membrane potential was mostly linear until a threshold number of spines was activated, at which often D-spikes along with supralinear summation set in. As to the mechanisms supporting active integration, NMDA receptors (NMDARs) strongly contributed to all aspects of supralinearity, followed by dendritic voltage-gated $Na^+$- and $Ca^{2+}$-channels, whereas local $Na^+$ spine spikes, as well as morphological variables, barely mattered.

Because of the low numbers of coactive spines required to trigger dendritic $Ca^{2+}$ signals and thus possibly lateral release of GABA onto mitral and tufted cells, we predict that thresholds for granule cell-mediated bulbar lateral inhibition are low. Moreover, D-spikes could provide a plausible substrate for granule cell-mediated gamma oscillations.

## Introduction

The classical role of dendrites is to receive synaptic or sensory inputs and to conduct the ensuing electrical signals toward the site of action potential initiation at the axon hillock. Because this conduction is passive for smaller membrane depolarizations, low numbers of coactive synaptic inputs are usually integrated in a linear fashion. However, the recent decades have revealed the presence of active dendritic conductances, most importantly, voltage-gated $Ca^{2+}$

other data that are not shown in any figure and the IGOR experiments that contain the modified cumulative data sets shown in S2 Fig are available from the Dryad database. According to Dryad, "Your dataset will now remain private until your related manuscript has been accepted. At that point we will begin the curation process. For private access during this review period, you may share your unpublished dataset using this temporary link: https://datadryad.org/stash/share/UCkmgZdnCAo eXdz9Gs3rMNpSzhDSKucIwXLLhTv-m5c. Your dataset has been assigned a unique identifier, called a DOI (doi:10.5061/dryad.mgqnk98wq). This DOI may be provided to your journal, but it will not work until your submission has been approved by Dryad curators."

**Funding:** This work was funded mainly by the German Federal Ministry for Education and Research (BMBF, 01GQ1104/01GQ1502), with additional equipment funding by LMU-Graduate School of Neuroscience (GSN), Deutsche Forschungsgemeinschaft (DFG)-SFB 870 and funding for staff from German Israeli Foundation (GIF; 1479-418.13; all to VE). The funders had no role in study design, data collection and analysis, decision to publish, or preparation of the manuscript.

**Competing interests:** The authors have declared that no competing interests exist.

**Abbreviations:** AMPAR, AMPA receptor; $Ca_v$, voltage-gated $Ca^{2+}$-channel; D-spike, dendritic $Na^+$-spike; DNI, 4-methoxy-5,7-dinitroindolinyl-L-glutamate trifluoroacetate; EPSP, excitatory postsynaptic potential; $Na_v$, voltage-gated $Na^+$-channel; NMDAR, NMDA receptor; OGB-1, Oregon Green BAPTA-1; O/I, output/input; sO/I, subthreshold O/I relationship; TTX, tetrodotoxin; uEPSP, uncaging-evoked EPSP.

and $Na^+$ channels ($Ca_v$s, $Na_v$s) and NMDA receptors (NMDARs) that can amplify locally suprathreshold electrical signals and thus, generate dendritic spikes in many neuron types. The onset of such spikes often results in supralinear summation with respect to the arithmetic sum of the individual synaptic potentials; dendritic $Na_v$s also facilitate backpropagation of axonal action potentials into the dendritic tree. Sublinear summation may also occur, depending on dendritic input impedance, the density of active conductances, and the distribution of synaptic inputs, both in the spatial and temporal domain [1, 2].

For example, cortical and hippocampal pyramidal cell dendrites are reported to feature modes of supralinear integration that are bolstered by the aforementioned active dendritic conductances and regenerative mechanisms associated with them, i.e., dendritic $Ca^{2+}$-spikes, dendritic $Na^+$-spikes (termed D-spikes in the following), and so-called NMDA-spikes [3–8]. Conversely, sublinear integration is performed, e.g., by GABAergic cerebellar stellate cell dendrites via reductions in driving force for large dendritic depolarizations [9].

Aside from such computations that ultimately convert analogue signals into binary code at the axon initial segment, using various modes of information processing [10], another functional outcome of dendritic integration is the release of transmitter from the dendrites themselves. Dendritic transmitter release occurs in many brain regions and is particularly well known from the retina and the olfactory bulb [11]. In the bulb, axonless inhibitory granule cells release GABA exclusively from spines on their apical dendrite that contain reciprocal dendrodendritic synapses with the excitatory mitral and tufted cells. Mitral and tufted cells do not communicate directly (unless they belong to the same glomerular unit and interact directly via their apical tufts [12, 13]). Rather, their only interaction happens via lateral inhibition mediated by granule cells and other local interneuron subtypes, of which granule cells are the most numerous [14]. Thus, the properties of dendritic integration in granule cells are essential for the onset and degree of lateral inhibition.

Dendritic excitability in granule cells already sets in with single-spine activation, because a single mitral/tufted cell input can trigger a local $Na^+$-spike within the spine [15]. This spine spike can cause reciprocal release of GABA via gating of high-voltage-activated $Ca_v$s [16]. Activation of larger numbers of spines is observed to result in global low-threshold $Ca^{2+}$-spikes, which are mediated by T-type $Ca_v$s [17–19]. Synaptically evoked dendritic $Na^+$-spikelets (D-spikes) have been reported from mouse, turtle, and frog granule cells, causing regional $Ca^{2+}$ entry [19–21]. Finally, full-blown global $Na^+$-spikes can be elicited by stimulation of a single glomerulus, resulting in substantial $Ca^{2+}$ entry throughout the granule cell dendrite, with larger amplitude and faster onset than $Ca^{2+}$ entry mediated by $Ca^{2+}$-spikes [22, 23].

So far, it is unknown how many coinciding mitral/tufted cell inputs are required to elicit these spike types—and therewith $Ca^{2+}$ entry also in nonactivated granule cell spines, possibly invoking lateral inhibition: If $Ca^{2+}$-spike-mediated $Ca^{2+}$ entry suffices to trigger lateral GABA release from at least some reciprocal spines, then the threshold for dendritic $Ca^{2+}$-spike generation is equivalent to the onset of lateral inhibition, whereas global $Na^+$-spikes are likely to cause lateral inhibition with greater efficiency. Pressler and Strowbridge [24] have predicted that at least 20 coactive mitral/tufted cell inputs (within a time window of 1 millisecond) are required to achieve global $Na^+$-spike generation with 50% reliability, in line with the rather hyperpolarized granule cell resting membrane potential $V_m = -80$ mV and median unitary excitatory postsynaptic potential (EPSP) amplitudes $\leq 2$ mV in our hands [15, 17].

Another intriguing question is whether the local spine $Na^+$-spikes can contribute to dendritic integration in granule cells. Is it conceivable that the spine spikes across an activated spine cluster can team up to ignite the local dendritic segment, resulting in a D-spike? Conventional sequential 2-photon uncaging of glutamate (which involves moving the 2D xy-scanner from one uncaging spot to the next) would preclude observations of such effects because of the

inactivation of $Na_v$s during the sequence. Therefore, we simultaneously activated spines in 3D with a holographic system [25]. Importantly, this paradigm is coherent with physiological activation, because the firing of mitral and tufted cells within a glomerular ensemble is precisely locked to the sniff phase and thus can be synchronized within 1 millisecond [26]. Holographic stimulation also enabled us to target sufficient numbers of inputs, a problem in 2D because of the low granule cell spine density (1–2 spines per 10 μm; [27]) and, indeed, allowed us to investigate the onset of nonlocal spiking and ultimately to elicit global $Na^+$-spikes.

## Results

To study synaptic integration within granule cell apical dendrites, we mimicked simultaneous mitral/tufted cell inputs to a defined number and arrangement of granule cell spines in the external plexiform layer by 2-photon uncaging of 4-methoxy-5,7-dinitroindolinyl-caged glutamate (DNI, [15, 28]) at multiple sites in 3D using a holographic projector [25]. Cells in juvenile rat acute brain slices were patch-clamped and filled with $Ca^{2+}$-sensitive dye Oregon Green BAPTA-1 (OGB-1, 100 μM) to record somatic $V_m$ and $Ca^{2+}$ influx into one or several stimulated spines and several dendritic locations by 2-photon $Ca^{2+}$ imaging within a 2D plane (see Materials and methods).

### Subthreshold dendritic integration

To characterize subthreshold dendritic integration in terms of somatic $V_m$, we first consecutively stimulated individual spines to obtain single-synapse uncaging-evoked EPSPs (single uEPSP), followed by simultaneous activation of the same spines, resulting in a compound uEPSP. The number of coactivated spines was increased until either the global $Na^+$-spike threshold or the available maximum were reached (10–12 spines, see Materials and methods). Under the given experimental conditions, we succeeded to elicit global $Na^+$-spikes in 34 out of 111 granule cells. In the representative granule cell in Fig 1A, 9 coactivated spines generated global $Na^+$-spikes in 4 out of 7 trials. This stochastic behavior at threshold was also observed in all other spiking cells in our sample. As to the number of global $Na^+$-spikes per response, 23 cells fired 1 spike at threshold, 6 cells fired doublets (e.g., Fig 1A), and 5 cells fired yet more spikes, with variations in spike numbers across trials in some cells. The average latency of the first spike was $42 \pm 40$ milliseconds (average $\pm$ SD); second spikes occurred at a mean latency of $86 \pm 70$ milliseconds from the first ($n = 11$). The average single uEPSP amplitude across all spiking granule cell spines was $1.4 \pm 1.4$ mV ($n = 272$ spines, distribution of individual uEPSP amplitudes, see S1A Fig). The integration of uEPSPs originating from several spines was quantified by comparing the amplitude of the arithmetic sum of the respective single uEPSP traces to the actually measured multispine compound uEPSP amplitude for increasing numbers of coactivated spines, yielding a subthreshold output–input relationship (sO/I) for each cell (reviewed in [1]).

The analysis of sO/Is (Fig 1B) indicates that (1) for low numbers of coactivated spines, the average sO/I relationship across cells was linear; (2) beyond a certain stimulation strength, the compound uEPSP amplitude exceeded the amplitude of the arithmetic single uEPSP sum by an output/input (O/I) ratio of at least 1.2 in the majority of cells ($n = 19$ of 29). We classified these sO/Is as supralinear. The choice of this criterion (O/I ratio $\geq 1.2$) is based on the large variance of single uEPSP amplitudes in our data set (see Materials and methods, S1 Fig). The number of cells classified as supralinear was found to be highly robust against a lowering of this criterion (see S1 Table). In these 19 cells, supralinearity was attained at an average of $6.7 \pm 2.6$ stimulated spines and always maintained beyond this threshold until global $Na^+$-spike generation (except for one cell where the last added single uEPSP was very large). (3)

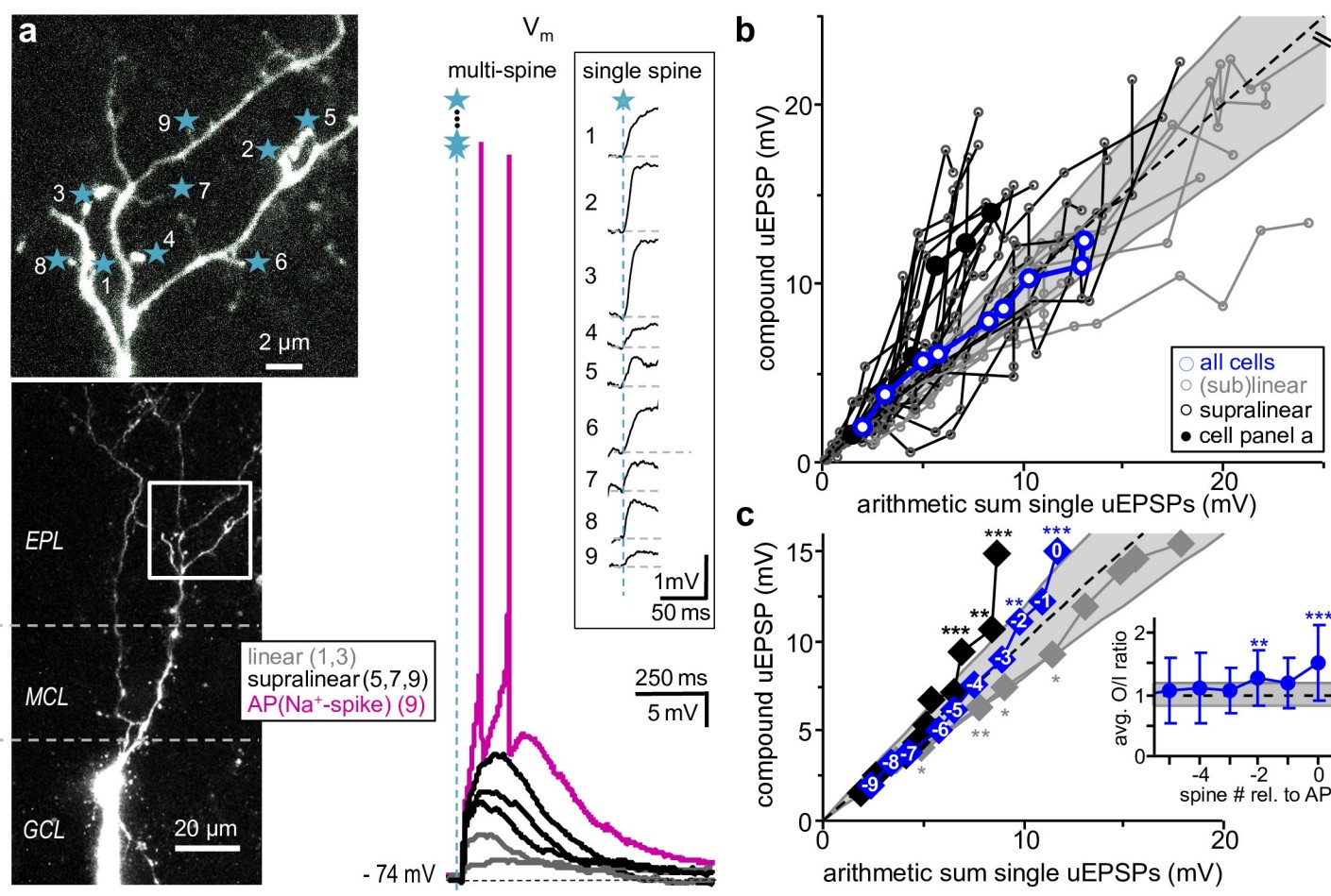

**Fig 1. Subthreshold dendritic integration in granule cells. a:** Left: Z-projection of 2-photon scan of representative cell, top part shows magnified inset with uncaging spots labeled by blue stars. Right: Somatic compound uEPSPs and global $Na^+$-spikes generated by simultaneous activation of 1, 3, 5, 7, and 9 spines (AP). Inset: Single uEPSPs recorded at the soma (see *Materials and methods*). **b:** sO/I of $n = 29$ individual experiments. Gray lines and circles ○: Sublinear to linear integration. Black lines and ○: Supralinear integration (solid circles: data from a). Blue lines and ○: Averaged sO/I of 1 to 9 coactivated spines across all GCs. Dashed line: linear y = x. Gray lines: Cut-off supra- and sublinear regime for classification of cells (y = 1.2x, y = 0.8x, see *Materials and methods*). 1 data point of 1 experiment exceeds the scale. **c:** sO/I cumulative plot of experiments in **b** with data arranged from -9 to 0 spines relative to global $Na^+$-spike threshold. Significance levels refer to O/I EPSP amplitude ratio distributions with means beyond the linear regime (0.8–1.2) tested against linearity (see inset, *Materials and methods*). Blue diamonds ♦: average sO/I of all experiments (see also inset): Supralinear at −2 ($p = 0.006$) and 0 spines ($p < 0.001$, mean O/I ratio 1.53 ± 0.63). Black diamonds ♦: average of supralinear sO/Is only ($n = 19$), significantly exceeding linear summation beyond −3 spines: −2 spines ($p < 0.001$), −1 spine ($p = 0.007$), 0 spine (i.e. at threshold, $p < 0.001$, mean O/I ratio 1.86 ± 0.52). Gray diamonds ♦: average of sublinear to linear sO/Is only ($n = 10$), significantly below linear summation below −3 spines: −7 spines ($p = 0.027$), −6 spines ($p = 0.008$), −5 spines ($p = 0.020$), −4 spines ($p = 0.021$, mean O/I ratio 0.79 ± 0.37). Inset: average O/I ratios of all experiments versus spine number relative to global $Na^+$-spike (AP) threshold. AP, action potential/global $Na^+$-spike; EPL, external plexiform layer; GCL, granule cell layer; MCL, mitral cell layer; O/I, output/input; sO/I, subthreshold O/I relationship; uEPSP, uncaging-evoked excitatory postsynaptic potential. In all figures, data means are presented ± standard deviation; $^*p < 0.05$, $^{**}p < 0.01$, $^{***}p < 0.001$.

Persistent sublinear integration (O/I ratio < 0.8) beyond a threshold was observed in only one cell, whereas the remaining 9 cells did not show any consistent deviations from linear behavior. In this subset of 10 cells, the average single uEPSP amplitude was significantly larger than for the 19 supralinear cells (2.1 ± 0.6 mV versus 1.1 ± 0.6 mV, $p < 0.001$).

Because each spiking granule cell required its individual spine number to reach the threshold for global $Na^+$-spike generation (for the respective stimulation pattern), we next aligned the sO/Is to the onset of the global $Na^+$-spike before averaging (Fig 1C; see Materials and methods). The ensuing averaged sO/I relationship was linear until global $Na^+$-spike threshold (corresponding to the number of coactivated spines that triggered a global $Na^+$-spike in a

subset of stimulations), where it turned supralinear. The averaged O/I ratios became significantly supralinear at −2 spines below threshold (see Fig 1C inset; see Materials and methods). For the averaged supralinear sO/Is (see above), the O/I ratio was highly significantly supralinear from −2 spines below threshold upwards. The average of the remaining linear/sublinear sO/Is was essentially linear, with a tendency toward sublinearity for lower numbers of coactivated spines. Thus, we find that dendritic $V_m$ integration is by and large linear at the granule cell soma, with a supralinear increase in $V_m$ close to global $Na^+$-spike threshold in the majority of cells.

## Transition from local spine spikes to nonlocal $Ca^{2+}$-spikes

Because granule cells are known to feature global $Ca^{2+}$-spikes and their generation had been associated with an increase in EPSP amplitude and duration [17], we investigated whether the onset of the supralinearity in somatic $V_m$ observed in the majority of sO/Is coincided with $Ca^{2+}$-spike generation. We detected the transition from local spine $Na^+$-spikes (which do not cause detectable dendritic calcium transients; [15, 17]) to $Ca^{2+}$-spike generation via two-photon $Ca^{2+}$ imaging in dendritic shafts that were on average 4.4 ± 3.3 μm remote from the base of the closest stimulated spine, thus not directly adjacent to the spines (e.g., Fig 2A; $n$ = 52 cells). Dendritic $Ca^{2+}$ transients were considered to indicate the presence of a $Ca^{2+}$-spike if their amplitude was above noise level ($\Delta F/F \geq 8\%$, see Materials and methods). We also always imaged at least one spine that was photostimulated throughout all spine combinations (termed spine 1 in the following), as exemplified in Fig 2A, showing somatic $V_m$ and concurrent $Ca^{2+}$ transients within spine 1 and at several dendritic locations with increasing numbers of stimulated spines. These dendritic $Ca^{2+}$ transients attenuated substantially while propagating from the activated spine set along the dendrite towards the soma (Fig 2B, $n$ = 38 locations in 12 cells). Thus, the $Ca^{2+}$-spike reported here is mostly a regional signal. Beyond the $Ca^{2+}$-spike threshold, higher numbers of activated spines resulted in larger dendritic $\Delta F/F$ signals with increased extent (Fig 2A and 2B), which can be explained by the recruitment of additional voltage-dependent conductances (see Results, Fig 7).

Across 28 granule cells that could produce both $Ca^{2+}$- and global $Na^+$-spikes under our experimental conditions, stimulation of, on average, 5.5 ± 2.1 spines sufficed for $Ca^{2+}$-spike generation (at an average somatic $V_m$ threshold of −67.8 ± 7.6 mV), whereas activation of 9.0 ± 1.6 spines was required to elicit a global $Na^+$-spike (at a somatic threshold of −60.2 ± 8.8 mV; both spine number and $V_m$ threshold: $p < 0.001$ $Ca^{2+}$- versus $Na^+$-spike, Fig 2C). In cells that did not yet fire a global $Na^+$-spike at the maximum number of stimulated spines, the threshold spine numbers for $Ca^{2+}$-spikes were not significantly different from those in spiking cells ($n$ = 25 analyzed cells; 5.3 ± 2.3 spines, respectively; $p$ = 0.82). The low somatic $V_m$ thresholds indicate distal initiation zones for both spike types.

Thus, $Ca^{2+}$-spike generation required substantially lower numbers of coactivated excitatory inputs than global $Na^+$-spike generation. However, when compound uEPSP properties were aligned to the $Ca^{2+}$-spike threshold spine number before averaging (Fig 2D, 2E and 2F), there was no discontinuous increase in amplitude or O/I ratio or kinetics at threshold (i.e., no significant difference from the linear fits to the subthreshold regime at threshold, see Materials and methods; see figure legend for $p$-values). Thus, the onset of a $Ca^{2+}$-spike as reported by dendritic $\Delta F/F$ is not substantially involved in the generation of $V_m$ supralinearity. The regional dendritic $Ca^{2+}$-spike observed here differs from earlier observations of granule cell global $Ca^{2+}$-spikes (also termed low-threshold spikes) that were generated by glomerular or external electrical field stimulation [17, 19], and that spread evenly throughout the dendrite and also boost and broaden somatic EPSPs (see Discussion).

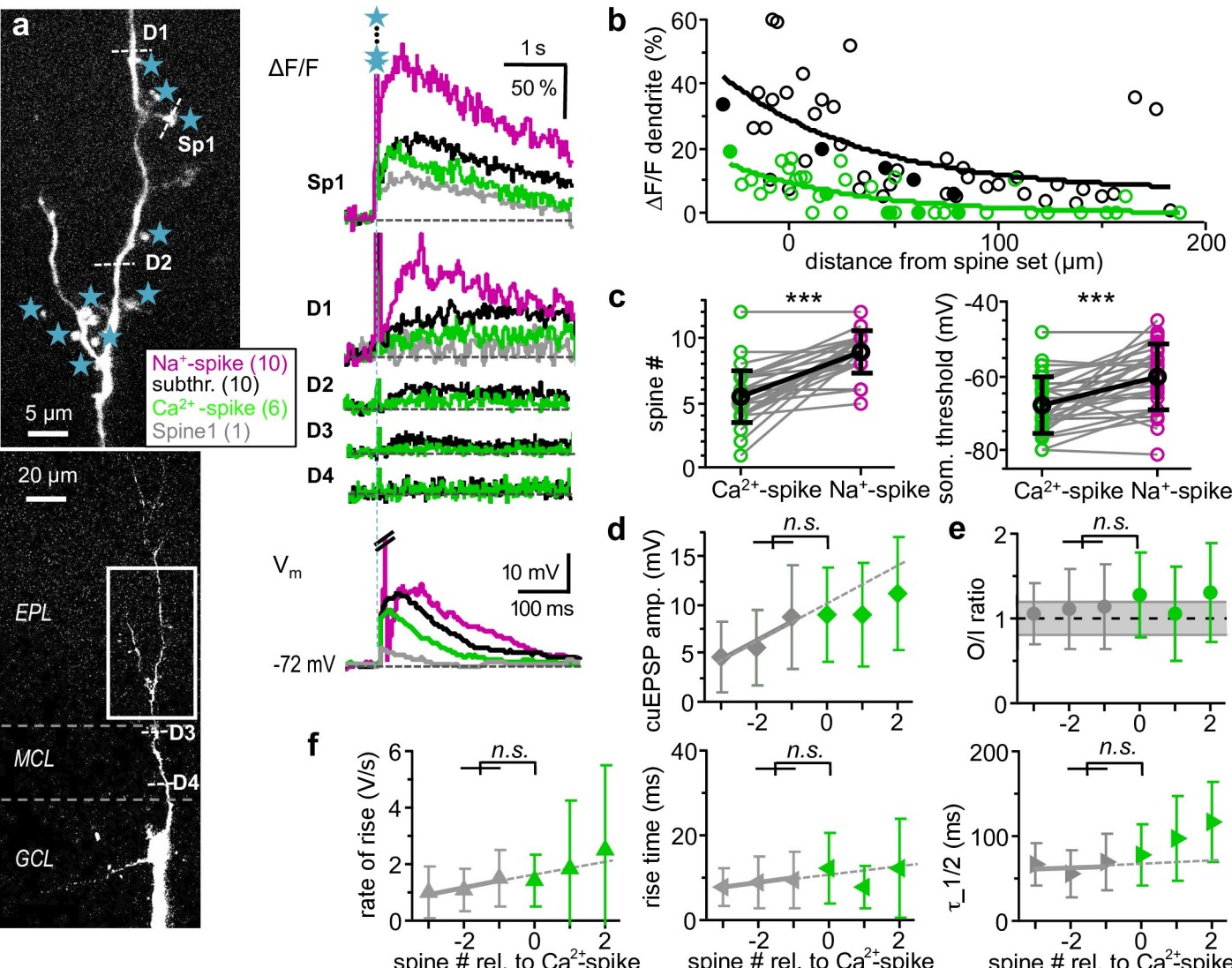

**Fig 2. Dendritic Ca²⁺-spikes: A nonlocal mode of dendritic activation. a:** Left: Scan of representative granule cell (as in Fig 1). Sp1 and D1-D4 indicate line scan sites, and stars indicate uncaging spots. Right, bottom: Somatic $V_m$ traces of single-spine and multisite uncaging (global Na⁺-spike truncated). Right, top: Averaged ΔF/F in the spine upon activation of 1, 6, 10, and 10 spines at the respective locations. Right, middle: Averaged ΔF/F in the dendrite measured at increasing distance from the activation site upon subthreshold activation of 6 and 10 spines. Truncated uncaging artefact in fluorescence traces. Gray: signals subthreshold for Ca²⁺-spike, green: at Ca²⁺-spike threshold, black: EPSPs and associated ΔF/F signals just subthreshold for global Na⁺-spike threshold, magenta: suprathreshold for global Na⁺-spike. **b:** Dendritic Ca²⁺ signals versus distance from the center of the stimulated spine set. Green circles ○: responses at Ca²⁺-spike threshold, black circles ○: responses for EPSPs just subthreshold for the global Na⁺-spike threshold or evoked by maximal available spine number. Data from 12 cells with ΔF/F data imaged at various distances from the set of stimulated spines. Solid symbols: Data from cell in **a**. Green and black lines: Exponential fits to respective data sets (at Ca²⁺-spike threshold: decay constant ± SD: λ = 61 ± 30 μm, ΔF/F(200 μm) = 0%, n = 38 data points; at or closer to global Na⁺-spike threshold: λ = 69 ± 47 μm, ΔF/F(200 μm) = 8%, n = 44 data points). **c:** Comparison of spine numbers (left) and somatic thresholds (right; both n = 28, p < 0.001, paired t-test) for Ca²⁺-spikes and global Na⁺-spikes. All error bars denote standard deviation, also in panels **d, e, f**. **d:** Mean somatic cuEPSP amplitudes with spine numbers aligned relative to Ca²⁺-spike threshold ([0]_Ca2+-spike; n = 25). Difference between [−2/−1] and [0] not significantly different from extrapolated linear fit (p = 0.29; Wilcoxon test, see *Materials and methods* for details of the test, see S2A Fig for data points from individual experiments). Gray symbols: subthreshold Ca²⁺-spike, green symbols: suprathreshold Ca²⁺-spike, dashed line: linear fit of subthreshold mean amplitudes, also for **e, f**. **e:** Mean O/I ratios aligned relative to Ca²⁺-spike, not significantly different from subthreshold (p = 0.78, n = 25). **f:** Kinetics of compound uEPSPs (n = 25 granule cells, see S2A Fig for data points from individual experiments): No significant increase above extrapolated linear fits at Ca²⁺-spike threshold for rate of rise (left, p = 0.52, n = 25) or rise time (middle, p = 0.49, n = 25) or half duration τ_1/2 (right, p = 0.42, n = 22). cuEPSP, compound uEPSP; EPL, external plexiform layer; GCL, granule cell layer; MCL, mitral cell layer; n.s., not significant; O/I, output/input; uEPSP, uncaging-evoked excitatory postsynaptic potential.

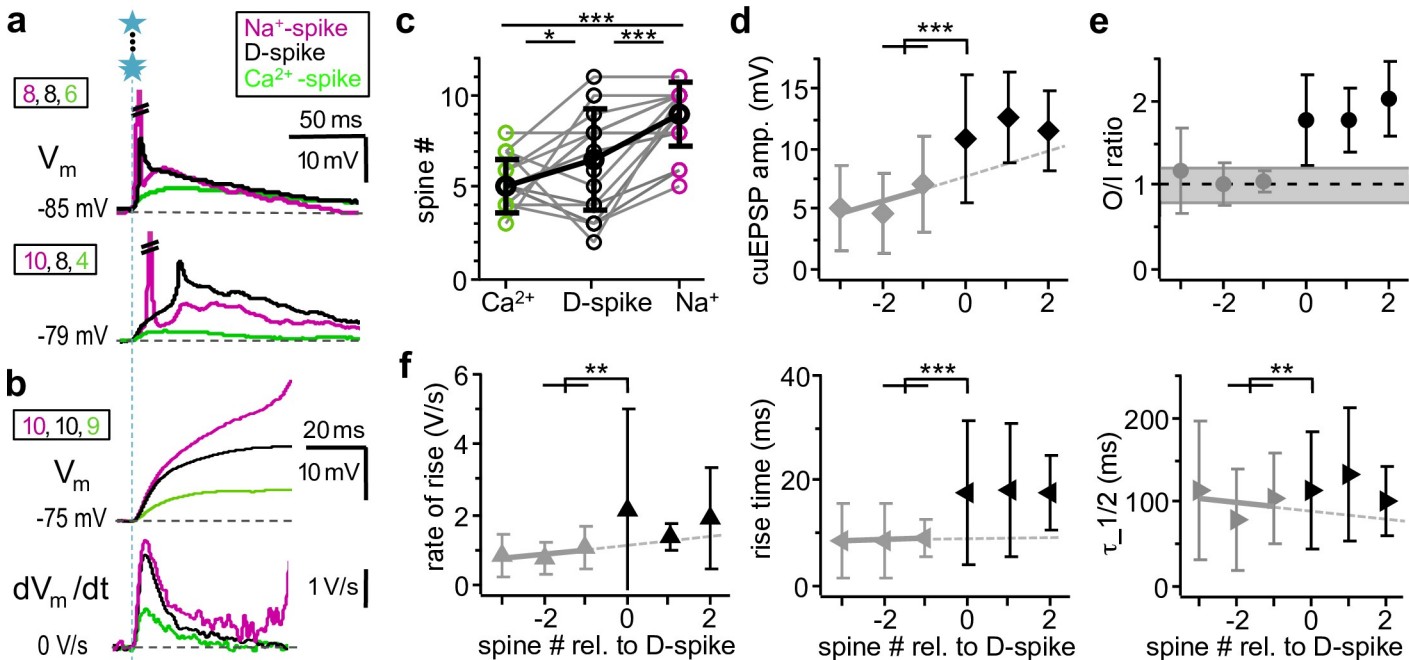

**Fig 3. D-spikes in approximately two thirds of granule cells: Spikelets and/or increased rate of rise associated with onset of supralinearity in O/I plot. a:** Examples of somatic spikelets recorded from 2 different cells at different numbers of coactivated spines. Green traces and spine numbers: Ca²⁺-spike threshold. Black traces and numbers: Na⁺-spikelets. Magenta traces and numbers: Full-blown global Na⁺-spikes, truncated (Na⁺-spike: short for global Na⁺-spike, also in **c**). **b:** Example cuEPSP $V_m$ recording in rising phase (top) and first derivative (bottom). Colors as in **a**, black here: D-spike, indicated by increase in rate of rise. **c:** Comparison of spine numbers sufficient to elicit Ca²⁺-spike, D-spike and global Na⁺-spike in the same granule cell ($n = 18$ cells; $F_{(2,53)} = 20.75$, $p < 0.001$, Ca²⁺-spike: $5.1 \pm 1.4$, D-spike: $6.5 \pm 2.7$, global Na⁺-spike: $8.9 \pm 1.7$ Holm–Sidak post hoc: Ca²⁺-spike versus D-spike: $p = 0.027$, Ca²⁺-spike versus global Na⁺-spike: $p < 0.001$, D-spike versus global Na⁺-spike: $p < 0.001$). **d:** Mean somatic cuEPSP amplitudes with spine numbers aligned relative to D-spike threshold for granule cells with supralinear sO/Is ($n = 18$; x-axis as in panel below). Difference between [−2/−1] and [0] highly significantly different from extrapolated linear fit ($p < 0.001$; Wilcoxon test, see *Materials and methods*, see S2B Fig for data points from individual experiments). Gray symbols: subthreshold D-spike, black symbols: suprathreshold D-spike, dashed line: linear fit of subthreshold mean amplitudes, also for **e, f**. **e:** Mean O/I ratios aligned relative to D-spike (x-axis as in panel below). Increase from $1.03 \pm 0.13$ to $1.77 \pm 0.54$, not tested, because the onset of supralinear O/I ratios was the criterion for the selection of cells and the setting of the D-spike threshold. **f:** Kinetics of cuEPSPs ($n = 18$ cells): Highly significant increases beyond extrapolated linear fits at D-spike threshold for rate of rise (left, $p = 0.002$, $n = 18$), rise time (middle, $p < 0.001$, $n = 18$) and half duration $\tau\_1/2$ (right, $p = 0.009$, $n = 16$). See S2B Fig for data points from individual experiments. Amp, amplitude; D-spike, dendritic Na⁺-spike; O/I, output/input; rel, relative; cuEPSP, compound uncaging-evoked excitatory postsynaptic potential.

## Transition to supralinear behavior due to D-spikes

Because the transition from linear to supralinear regimes in the cells with supralinear sO/Is could not be explained by the onset of Ca²⁺-spikes, we next investigated this transition in greater detail (in the cells with supralinear sO/Is, $n = 18$). We noticed that this transition was always linked to the occurrence of spikelets (e.g., Fig 3A) and/or an increase in the maximal rate of rise (e.g., Fig 3B), which are both known to indicate D-spikes ([3, 20, 29–32]). Spikelets were observed in 7 out of the 18 cells. The transition to D-spiking happened at an average of $6.5 \pm 2.7$ spines, significantly higher than the spine number required for Ca²⁺-spike onset and lower than for Na⁺-spikes (Fig 3C). Arrangement of the data relative to the transition spine number (Fig 3D) shows a highly significant discontinuous increase in compound uEPSP amplitudes at threshold (i.e., significantly different from linear fit to subthreshold regime, see Materials and methods), and the concomitant increase in O/I ratios (Fig 3E). This alignment also revealed highly significant increases of several compound uEPSP kinetic parameters, namely, maximal rate of rise, rise time, and half duration (Fig 3F).

Although the increased maximal rate of rise—as mentioned previously—is a hallmark of D-spikes, how can D-spikes be consistent with the observed increase in compound uEPSP rise

time? Previously, we had observed that single uEPSP rise time even increased upon $Na_v$ blockade, because of the block of the spine $Na^+$-spike [15].

This apparent discrepancy can be explained by the substantial latency of D-spikes, as evident from the latency between uncaging onset and the peak of spikelets, which was $21 \pm 19$ milliseconds (median 10 milliseconds; $n = 7$ cells; see Fig 3A bottom and Discussion). Furthermore, our pharmacological experiments (see next) prove that the observed changes in compound uEPSP kinetics in supralinear cells were indeed due to the activation of dendritic $Na_v$s.

In conclusion, the supralinearity observed in the sO/Is of the majority of granule cells is due to the onset of a D-spike.

## Additional $Ca^{2+}$ influxes into the spine mediated by $Ca^{2+}$-, D-, and global $Na^+$-spike

Next, we asked whether dendritic signals such as $Ca^{2+}$-spikes, D-spikes, and global $Na^+$-spikes can boost $Ca^{2+}$ influx into spines that are already activated by local inputs. Such summation had been observed previously for both synaptically evoked global $Na^+$- and $Ca^{2+}$-spikes [17, 22].

An exemplary transition from local spine activation to $Ca^{2+}$-spike to D-spike to full-blown $Na^+$-spike is shown in Fig 4A, and in Fig 4B and 4C, all normalized $Ca^{2+}$ signals are arranged relative to global $Na^+$-spike threshold in the imaged spine 1 (number 1 with respect to the entire set of spines) and dendrite. Because for low numbers of coactivated spines (1–4, not aligned to any threshold), there was no significant difference in the spine 1 $Ca^{2+}$ signal, we normalized $\Delta F/F$ for each spine 1 to its mean of (1–4) to reduce variance (see Materials and methods). From 5 coactive spines below global $Na^+$-spike threshold onwards, both average spine 1 and dendritic $\Delta F/F$ increased continuously.

Arrangement of the data relative to the $Ca^{2+}$-spike threshold spine number $[0]_{Ca2+\text{-spike}}$ (as detected in the dendrite, Fig 4E left; note that at $[0]_{Ca2+\text{-spike}}$, there was always a dendritic $Ca^{2+}$ signal, different from the global $Na^+$-spike threshold, where EPSPs and $Na^+$-spikes occurred stochastically) revealed that below threshold spine 1 $\Delta F/F$ was rather constant, whereas at threshold, a highly significant increase in $\Delta Ca^{2+}$ occurred (by, on average, $\pm$SD: $1.44 \pm 0.80$, $[0]_{Ca2+\text{-spike}}$ versus $[-1/-2]_{Ca2+\text{-spike}}$, $n = 26$ spines, Fig 4D left). Similarly, arrangement of the data relative to the D-spike threshold spine number $[0]_{D\text{-spike}}$ also revealed a highly significant step-like increase in spine 1 $\Delta F/F$ (by $1.75 \pm 0.85$, $[0]_{D\text{-spike}}$ versus $[-1/-2]_{D\text{-spike}}$, $n = 18$ spines, Fig 4D right) and a significant increase in dendritic $\Delta F/F$ (by $1.76 \pm 0.74$, $[0]_{D\text{-spike}}$ versus $[-1/-2]_{D\text{-spike}}$, $n = 9$, Fig 4E right; again, these changes always occurred at $[0]_{D\text{-spike}}$ with no stochastic variation). Finally, global $Na^+$-spike generation lead to yet more substantial, highly significant additional $Ca^{2+}$ influx into both the spine ($2.03 \pm 1.11$ increase for $Na^+$-spike versus EPSP at the global $Na^+$-spike threshold spine number $[0]_{Na+\text{-spike}}$, absolute $84\% \pm 59\%$ $\Delta F/F$, $n = 18$ spines, Fig 4B) and the dendrite ($2.03 \pm 1.12$ increase for $Na^+$-spike versus EPSP at $[0]_{Na+\text{-spike}}$, absolute $41\% \pm 20\%$ $\Delta F/F$, $n = 11$, Fig 4C). Compared with the local synaptic input and its ensuing spine $Na^+$-spike, global $Na^+$-spikes increased spine $Ca^{2+}$ entry by $3.08 \pm 1.32$ (Fig 4B), thus coincident local inputs and global $Na^+$-spikes summate highly supralinearly (see Discussion).

We infer that all 3 types of nonlocal signals, $Ca^{2+}$-spike, D-spike, and global $Na^+$-spikes, can mediate substantial additional $Ca^{2+}$ influx into the spine on top of the contribution of the local synaptic input. Thus, a granule cell spine "knows" about its parent dendrite's general excitation level. Similar step-like increases between nonlocal signals will occur in dendrites close to the activated spine set and also in nearby silent spines (not receiving direct inputs, not investigated here), because those were found previously to respond with similar $\Delta Ca^{2+}$ to nonlocal spikes as dendrites [17, 22, 33].

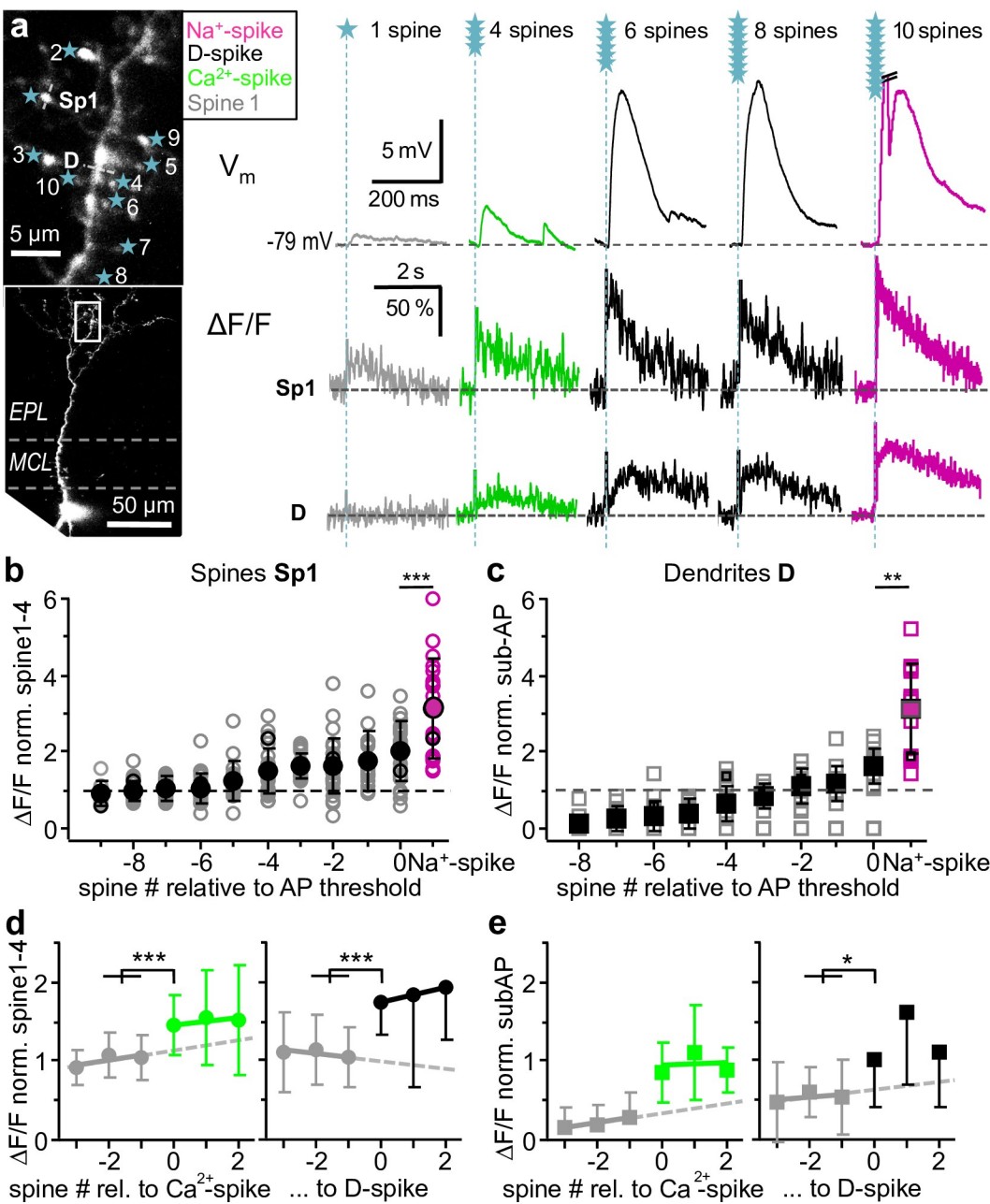

**Fig 4. Additional ΔCa²⁺ in spine and dendrite caused by nonlocal spikes. a:** Left: Scan of representative granule cell. Sp1 and D indicate line scan sites, stars indicate uncaging spots. Right: Somatic $V_m$ recordings (top) and line scans in spine (Sp1) and dendritic location (D) for increasing spine numbers (bottom). Gray traces: below Ca²⁺-spike, green traces: at Ca²⁺-spike threshold; black traces: at and above D-spike threshold; magenta traces: suprathreshold for global Na⁺-AP. Na⁺-AP truncated for clarity. **b:** Spine Ca²⁺ signals ΔF/F normalized to Sp1-4 (see *Materials and methods*) and aligned to Na⁺-AP threshold ($n$ = 33 spines in 16 GCs); Gray circles ∘: individual spines, solid black •: mean; open black ∘: spine from **a**, open magenta ∘: AP data from individual spines; solid magenta with black outline •: mean global Na⁺-spike data (responses with more than 1 AP were not taken into account). Gradual increase from −6 spines onwards of spine ΔF/F with highly significant additional increase upon AP generation ($n$ = 20 pairs, Wilcoxon, $p$ < 0.001). **c:** Dendritic Ca²⁺ signals ΔF/F normalized to mean above Ca²⁺-spike threshold and below global Na⁺-spike threshold (see *Materials and methods*; $n$ = 19 cells). Symbols as in right panel, with squares instead of circles. Gradual increase from −5 spines onwards with significant additional increase upon global Na⁺-spike generation ($n$ = 12 pairs, Wilcoxon, $p$ = 0.001). **d:** Spine ΔF/F normalized as in b, arranged relative to Ca²⁺-spike threshold (left panel, $n$ = 26 in 14 cells, Wilcoxon test, $p$ < 0.001, see *Materials and methods*) and relative to D-spike threshold (right panel, $n$ = 19 in 10 cells, $p$ < 0.001). See S2A and S2B Fig for individual data points. Gray symbols: subthreshold Ca²⁺-spike/D-spike, green symbols: suprathreshold Ca²⁺-spike, black symbols: suprathreshold D-spike, dashed

line: linear fit of subthreshold mean amplitudes, also for **e**. **e**: Dendrite $\Delta F/F$ normalized as in c, arranged relative to $Ca^{2+}$-spike threshold (left panel, $n = 17$, significance not tested, because increase in dendritic $\Delta F/F$ above noise level was criterion for onset of $Ca^{2+}$-spike) and relative to D-spike threshold (right panel, $n = 12$, Wilcoxon, $p = 0.015$, see S2B Fig for individual data points). Symbols as in **d**, with squares instead of circles. AP, action potential; D-spike, dendritic $Na^+$-spike.

## Molecular mechanisms of integration: Na$_v$s

We observed previously [15] that single-granule cell spine activation resulted in a local $Na_v$-dependent spine spike. Although most of the postsynaptic $Ca^{2+}$ entry was mediated by NMDARs that were unblocked already by the AMPA receptor (AMPAR)-mediated EPSP, the spine spike contributed additional $Ca^{2+}$ by gating of high-voltage-activated $Ca_v$s. Notably, somatically recorded single uEPSPs were not reduced in amplitude by $Na_v$ blockade, but slowed down, indicative of a strong filtering effect by the spine neck and possibly the dendrite [15]. Nevertheless, could spine spikes themselves eventually occur simultaneously across a few clustered spines and thus engender nonlocal spiking?

For all pharmacological interventions related to dendritic integration mechanisms below the global $Na^+$-spike threshold, we stimulated 1, 2, 4, 6, 8, and 10 spines before and after wash-in of the drug (see Materials and methods). We blocked $Na_v$s by wash-in of 0.5–1 μM tetrodotoxin (TTX, $n = 12$ cells). Amplitudes of single and compound uEPSPs were unaltered (Fig 5A and 5B). However, the significant increase of average O/I ratios from 6 to 8 coactivated spines in control was blocked in the presence of TTX (Fig 5C). 4 of the 12 cells fired a global $Na^+$-spike upon stimulation of 10 spines, which was always abolished by wash-in of TTX.

Across all 12 cells, the spine 1 $Ca^{2+}$ signal and dendritic $Ca^{2+}$ signals were significantly reduced in TTX for all numbers of activated spines (Fig 5D).

In 7 out of these 12 cells summation was supralinear and thus, as shown above, associated with the occurrence of D-spikes. In this set of cells, TTX application significantly reduced both the increases in compound uEPSP rise time and maximal rate of rise at supralinearity threshold (Fig 5E and 5F). Note that below threshold compound, uEPSP rise times were indeed slowed in TTX, in line with our previous observations on single uEPSPs [15].

Moreover, the significant increase in $\Delta Ca^{2+}$ within activated spines associated with the transition to the D-spike (see Fig 4D) was also sensitive to $Na_v$ blockade in these experiments (5G and 5H). This observation further proves the presence of a $Na_v$-mediated D-spike, because dendritic $Na_v$ activation will recruit both low- and high-voltage-activated $Ca_v$s, further augmenting $Ca^{2+}$-spikes [33, 34].

In summary, $Na_v$ blockade had only subtle but significant effects on somatic $V_m$ summation on average (see Discussion). Dendritic $Na_v$ activation, however, underlies the D-spike and the additional $Ca^{2+}$ entry into spines associated with it.

## Molecular mechanisms of integration: Key role of NMDARs

NMDARs have been shown to contribute substantially to local postsynaptic signaling in granule cells [15, 17, 35] and to foster the generation of global $Ca^{2+}$-spikes [17]. NMDARs are also known to boost dendritic integration in cortical pyramidal cells (via so-called NMDA-spikes; [4]).

To investigate the contribution of NMDARs to dendritic integration, we blocked NMDARs by wash-in of APV (25 μM) in $n = 8$ experiments (Fig 6A). The compound uEPSP amplitude was substantially reduced from 4 activated spines onwards (Fig 6B). Although under control conditions, we observed supralinear integration from 4 spines onwards, blocking of NMDARs switched the average sO/I relationship to linear integration (Fig 6C). In 2 experiments, cells fired a global $Na^+$-spike upon stimulation of 10 spines under control conditions, and in one of

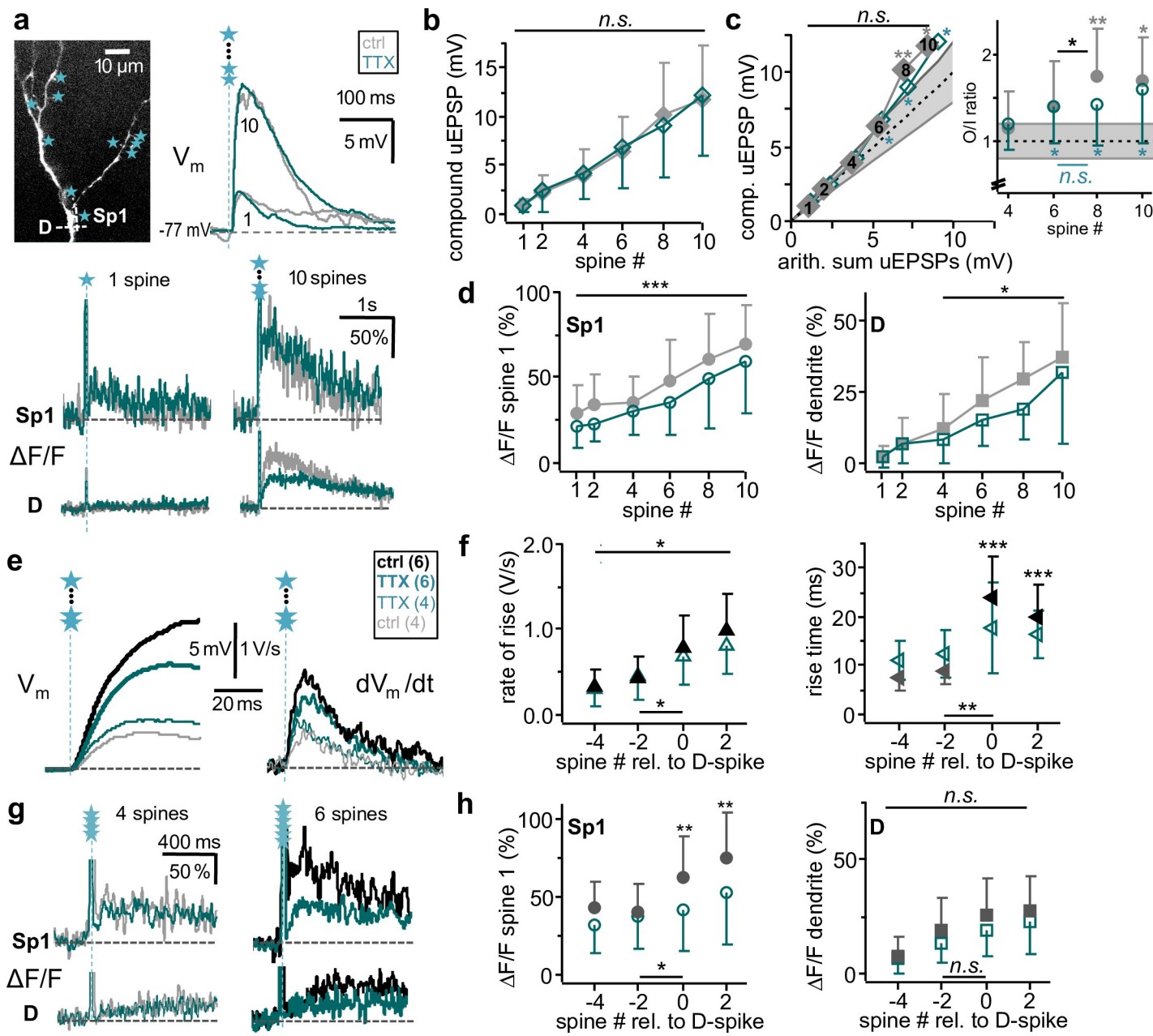

**Fig 5. Molecular mechanisms of subthreshold integration: Na$_v$s. a:** Example Na$_v$ blockade experiment. Top left: Stimulated spine set with line scan sites Sp1, D, and uncaging spots. Top right: Somatic V$_m$ recording of compound uEPSPs, below spine Sp1 and dendrite D ΔF/F for 1 and 10 spines. Gray traces: control. Blue traces: TTX (0.5–1 μM). **b:** Effect ($n$ = 12 cells) of 0.5–1 μM TTX on somatic compound uEPSP amplitude upon activation of 1, 2, 4, 6, 8, and 10 spines. Repeated measures 2-way ANOVA (Materials and methods, also in d,f, h): no interaction effect (spine # x TTX): F$_{(5,119)}$ = 2.54, $p$ = 0.091; no TTX effect: F$_{(1,119)}$ = 0.97, $p$ = 0.352. **c:** Cumulative effect ($n$ = 12 cells) of Na$_v$ blockade on averaged sO/I. No interaction effect (spine # x TTX): F$_{(5,99)}$ = 1.60, $p$ = 0.195; no TTX effect: F$_{(1,99)}$ = 0.84, $p$ = 0.385 (n.s. above). Lines and grey asterisks as in Fig 1C. The average O/I ratio for 8 spines was highly significantly supralinear in control (** above data points, $p$ < 0.01) and significantly increased versus 6 spines (* in inset, $p$ < 0.05, Wilcoxon test). In TTX, O/I ratios were still significantly supralinear (* below data points, $p$ < 0.05 for all), but the increase from 6 to 8 spines disappeared (inset, n.s. below). **d:** Effect of Na$_v$ blockade on average ΔF/F in spine 1 (Sp1, left, $n$ = 25 spines in 11 cells) and dendrites (D, right, $n$ = 12) upon activation of 1–10 spines. No interaction effect on spine 1 ΔF/F (spine # x TTX): F$_{(5,239)}$ = 1.69, $p$ = 0.145; TTX effect: F$_{(1,239)}$ = 15.16, average reduction to 0.89 ± 0.54 of control, $p$ < 0.001. No interaction effect on dendrite ΔF/F from 4 spines onwards (spine # x TTX): F$_{(3,79)}$ = 0.46, $p$ = 0.715; TTX effect: F$_{(1,79)}$ = 9.29, average reduction to 0.75 ± 0.28 of control, $p$ = 0.014. Asterisks above error bars: significance of difference TTX versus control. **e:** Example for effect of TTX on compound uEPSP kinetics below and at D-spike threshold (4 and 6 coactivated spines in this granule cell, respectively). Left traces: V$_m$, right traces: dV$_m$/dt. Gray/black traces: control. Blue traces: TTX (0.5–1 μM). Thin lines: 4 spines, thick lines: 6 spines. Note the reduction in maximal rate of rise above threshold but not subthreshold. **f:** Cumulative data for effect of TTX ($n$ = 7 cells with supralinear sO/Is) on compound uEPSP rate of rise (left) and rise time (right). Repeated measures two-way ANOVA (see Materials and methods, also below): no interaction effect on rate of rise (spine # x TTX): F$_{(3,55)}$ = 3.06, $p$ = 0.055; TTX effect: F$_{(1,55)}$ = 8.25, $p$ = 0.028. Interaction effect on rise time (spine # x TTX): F$_{(3,55)}$ = 12.49, $p$ < 0.001. Asterisks indicate significance of differences between TTX and control (* $p$ = 0.028, *** $p$ <

0.001). Asterisks at bottom indicate significance of differences of parameter increases from -2 to 0 between control and TTX (Wilcoxon test; rate of rise: $p < 0.05$ [$W = 17$, $n_{sr} = 6$], rise time: $p < 0.01$ [$W = 28$, $n_{sr} = 7$]). See S2C Fig for individual data points. **g:** Example for effect of TTX on spine 1 and dendrite ΔF/F below and above D-spike threshold (4 and 6 coactivated spines, respectively; same cell as in **e**, same color code). Note the reduction in spine 1 ΔF/F by TTX at threshold but not subthreshold. **h:** Cumulative data for effect of TTX ($n = 7$ cells with D-spike) on ΔF/F in spines (left, $n = 13$) and dendrite (right, $n = 7$). Full symbols: control, open symbols: TTX. Repeated measures 2-way ANOVA (see Materials and methods): interaction effect on spine ΔF/F (spine # x TTX): $F_{(3,103)} = 3.20$, $p = 0.035$. No interaction effect on dendrite ΔF/F: $F_{(3,55)} = 0.66$, $p = 0.588$; no TTX effect: $F_{(1,55)} = 5.10$, $p = 0.065$. Asterisks above indicate significance of differences between TTX and control. Asterisks at bottom indicate significance of differences of ΔF/F from −2 to 0 between control and TTX (Wilcoxon test; spine S: $p = 0.029$ [$W = 55$, $n_{sr} = 13$], dendrite: not significant [$W = −2$, $n_{sr} = 5$]). See S2C Fig for individual data points. arith., arithmetic; D-spike, dendritic Na$^+$-spike; n.s., not significant; sO/I, subthreshold output/input; TTX, tetrodotoxin; uEPSP, uncaging-evoked excitatory postsynaptic potential.

these, there were somatic spikelets upon stimulation of 8 and 10 spines. All were abolished by wash-in of APV.

APV also highly significantly reduced spine 1 Ca$^{2+}$ signals for all stimulation strengths, effectively blocking the linear control increase in ΔCa$^{2+}$ (e.g., at 8 costimulated spines, spine ΔF/F: 0.25 ± 0.18 of control, $p < 0.001$; Fig 6D). Moreover, APV strongly reduced dendritic ΔCa$^{2+}$ and thus prevented Ca$^{2+}$-spike generation (Fig 6D; e.g., at 8 spines, dendrite ΔF/F: 0.14 ± 0.15 of control, $p = 0.003$). APV reduced the half duration of compound uEPSPs from 4 spines onwards (interaction effect [spine # x APV]: $F_{(5,95)} = 3.20$, $p = 0.017$, absolute mean values at 8 coactivated spines: $\tau\_1/2$ control 90 ± 53 milliseconds, APV 37 ± 21 milliseconds, see data repository) but did not interfere with fast kinetics, e.g., the maximal rate of rise of the compound uEPSP (no interaction effect [spine # x APV]: $F_{(5,95)} = 1.62$, $p = 0.182$; no APV effect: $F_{(1,95)} = 2.26$, $p = 0.176$, $n = 8$ cells, see data repository).

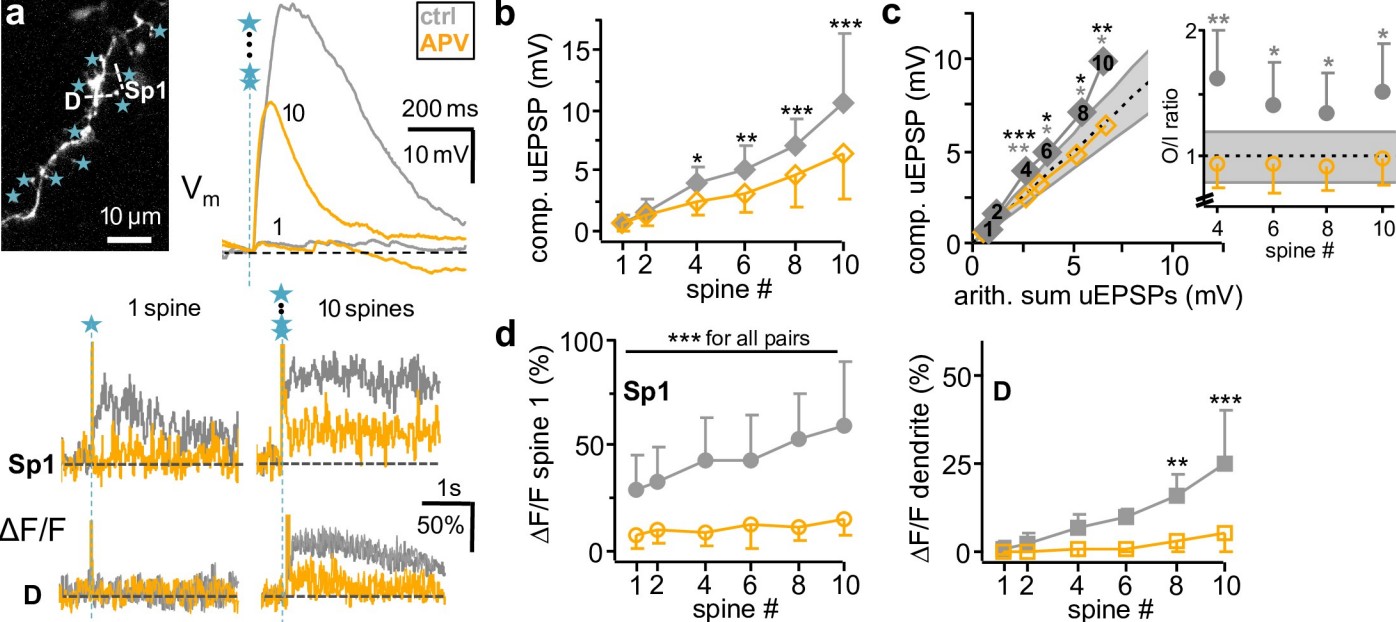

**Fig 6. Molecular mechanisms of subthreshold integration: NMDA receptors. a:** Example NMDAR blockade experiment with strong NMDAR component. Top left: Stimulated spine set with line scan sites Sp1, D, and uncaging spots. Top right: Somatic V$_m$ recording of uEPSP, below spine (Sp1), and dendrite (D) ΔF/F for 1 and 10 spines. Gray traces: control. Yellow traces: APV (25 μM). **b:** Cumulative effect ($n = 8$ cells) of 25 μM APV on somatic cuEPSP amplitude upon activation of 1, 2, 4, 6, 8, and 10 spines. Interaction effect (spine # x APV): $F_{(5,95)} = 8.08$, $p < 0.001$. Black asterisks: significance of difference APV versus control. **c:** Effect of NMDAR blockade on averaged sO/I upon activation of 1–10 spines. Interaction effect (spine # x APV): $F_{(5,95)} = 3.37$, $p = 0.014$, $n = 8$. Black asterisks above data points: significance of difference APV versus control. Lines and grey asterisks as in Fig 1C. Control O/I ratios from 4 spines upwards were supralinear, which all became linear in APV. **d:** Effect of NMDAR blockade on average ΔF/F in Sp1 (left, $n = 15$ spines) and D (right, $n = 8$) upon activation of 1–10 spines. Interaction effect on spine ΔF/F (spine # x APV): $F_{(5,179)} = 6.36$; $p < 0.001$. Interaction effect on dendrite ΔF/F (spine # x APV): $F_{(5,95)} = 8.34$, $p < 0.001$. Asterisks: significance of difference APV versus control; arith., arithmetic; comp., compound; cuEPSP, compound uncaging-evoked excitatory postsynaptic potential; sO/I, subthreshold output/input.

Thus, on top of the strong NMDAR contribution to local postsynaptic $Ca^{2+}$ entry, all types of nonlocal granule cell spikes and their associated $Ca^{2+}$ influxes are highly NMDAR-dependent, even though NMDAR activation happens in the electrically isolated spine heads (see Discussion).

## Molecular mechanisms of integration: Contribution of both low- and high-voltage-activated $Ca_v$s to dendritic $Ca^{2+}$ entry

To verify whether distally evoked $Ca^{2+}$-spikes in granule cell dendrites are mediated by T-type $Ca_v$s as observed earlier for global $Ca^{2+}$-spikes evoked by glomerular stimulation [17], we investigated their contribution to multispine signals in $n = 11$ cells (Fig 7A). Wash-in of 10 μM mibefradil ($IC_{50}$: T-type $Ca_v$s 2.7 μM, L-type $Ca_v$s 18.6 μM [36]) did not alter compound uEPSPs upon activation of up to 8 spines. For 10 spines, compound uEPSPs were slightly but significantly reduced by on average $0.8 \pm 1.4$ mV ($p = 0.01$, Fig 7B). Coactivation of 10 spines also lead to supralinear $V_m$ summation in control (Fig 7C), which was reduced by blockade of T-type $Ca_v$s. Compound uEPSP kinetics were unaltered (see data repository). In one experiment, a global $Na^+$-spike was generated upon stimulation of 10 spines under control conditions, which was abolished by mibefradil.

$Ca^{2+}$ signals in spine 1 and dendrites were significantly reduced for all spine numbers (spine 1: average $\pm$ SD $0.74 \pm 0.31$ of control $\Delta F/F$, $p < 0.001$; dendrite: $0.74 \pm 0.38$ of control, $p = 0.003$, Fig 7D). However, mibefradil did not entirely block dendritic $\Delta Ca^{2+}$ upon stimulation of 4 spines and beyond (remaining signal $16 \pm 9\%$ $\Delta F/F$ at 10 coactivated spines).

To identify the source for the remaining dendritic $\Delta F/F$, we additionally washed in 100 μM $Cd^{2+}$ to block high-voltage-activated $Ca_v$s [34] in a subset of 4 cells (Fig 7E–7G). $Cd^{2+}$ effectively abolished the dendritic $Ca^{2+}$ signal and substantially further reduced the spine 1 $Ca^{2+}$ signal to $0.52 \pm 0.26$ of mibefradil or $0.41 \pm 0.22$ of control ($n = 8$ spines), leaving the compound uEPSP unaltered.

We conclude that T-type $Ca_v$s substantially contribute to $Ca^{2+}$ entry into the spine and dendrite during dendritic integration and mediate the onset of the $Ca^{2+}$-spike, but that high-voltage-activated $Ca_v$s also contribute, most likely involving additional $Ca^{2+}$ entry via L-type $Ca_v$s or other channel types that are activated by D-spikes. Both low- and high-voltage-activated $Ca_v$s did not substantially influence somatic $\Delta V_m$ in our stimulation paradigm.

## Limited influence of morphology on nonlocal spike generation

To determine whether the spacing of stimulated spines, the average spine neck length and other morphological variables influenced the efficacy of activated subsets of spines to elicit nonlocal spiking, we analyzed the positions of the stimulated spines relative to the granule cells' dendritic tree as reconstructed in 3D and checked for correlations (Fig 8A–8E, see Materials and methods). Table 1 shows that only 2 out of 12 variables correlated with $Ca^{2+}$-spike in terms of coactivated spine numbers, whereas both D-spike and global $Na^+$-spike initiation threshold spine numbers did not correlate significantly with any tested variable, with a weak trend for a positive correlation between spine distribution and global $Na^+$-spike initiation (Fig 8C). $Ca^{2+}$-spike generation was facilitated by close packing of spines that were located on the same and/or a rather low number of branches (Fig 8C and 8E). Finally, developmental effects might influence synaptic density and excitability in early born granule cells within the age range used here [37, 38]; however, there were no correlations between threshold spine numbers and animal age (Fig 8F).

Within the experimentally accessible range of variables, individual spine sets have, by and large, an equal impact on local and global $Na^+$-spike generation, independent from granule

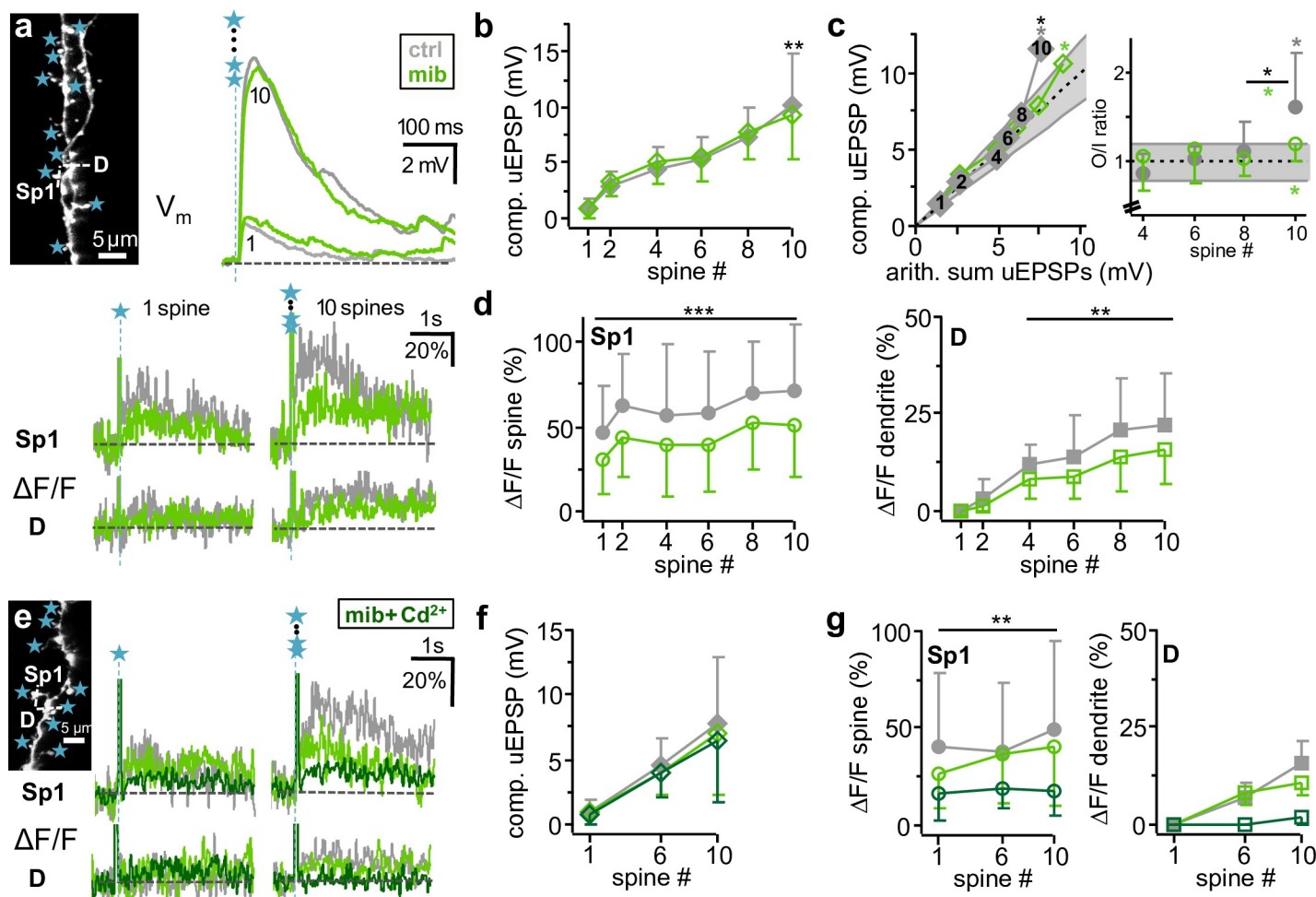

**Fig 7. Molecular mechanisms of subthreshold integration: Low- and high-voltage-activated Ca$_v$s. a:** Example low-voltage-activated Ca$_v$ blockade experiment. Top left: Stimulated spine set with line scan sites Sp1, D, and uncaging spots. Top right: Somatic V$_m$ recording of uEPSP, below spine (Sp1) and dendrite (D) ΔF/F for 1 and 10 spines. Grey traces: control. Green traces: mib, 10 μM. **b:** Cumulative effect ($n = 11$ cells) of mib on somatic compound uEPSP amplitude upon activation of 1, 2, 4, 6, 8, and 10 spines. Repeated measures 2-way ANOVA (see *Materials and methods*): interaction effect (spine # x mib) F$_{(5,131)} = 3.88$, $p = 0.010$. Asterisks: significance of difference mib versus control. **c:** Cumulative effect ($n = 8$ cells) of LVA Ca$_v$ blockade on averaged sO/I upon activation of 1–10 spines. Interaction effect (spine # × mib) F$_{(4,69)} = 4.69$, $p = 0.006$. Lines as in Fig 1C. Black asterisks above error bars indicate significance of differences between mib and control. Gray and green asterisks/ significance levels above error bars refer to O/I ratio distributions with means beyond the linear regime (0.8–1.2) tested against linearity (*$p < 0.05$, as in Fig 1C). Integration was significantly supralinear at 10 spines for both control and mib. The increase in O/I ratios between 8 and 10 spines was also significant but significantly smaller in mib versus control (inset *, Wilcoxon test). **d:** Effect of LVA Ca$_v$ blockade on average ΔF/F in Sp1 (left, $n = 26$ spines in 11 cells) and D (right, $n = 11$) upon activation of 1–10 spines. No interaction effect on spine ΔF/F (spine # x mib): F$_{(5,311)} = 0.26$, $p = 0.933$; mib effect: F$_{(1,311)} = 60.16$, $p < 0.001$. No interaction effect on dendrite ΔF/F (spine # x mib): F$_{(3,87)} = 1.11$, $p = 0.359$; mib effect: F$_{(1,87)} = 15.84$, $p = 0.003$. Asterisks: significance of difference mib versus control. **e:** Example for subsequent blockade of low- and high-voltage-activated Ca$_v$s on ΔF/F in spine (Sp1) and dendrite (D) for 1 and 10 spines. Top left inset: Scan of stimulated spine set with indicated line scan sites Sp1, D and uncaging spots. Gray traces: Control. Green traces: mib (10 μM). Dark green: added Cd$^{2+}$ (100 μM). **f:** Effect of subsequent low- and high-voltage-activated Ca$_v$ blockade on somatic compound uEPSP amplitude upon activation of 1, 6, and 10 spines ($n = 4$ cells). **g:** Effect of subsequent low- and high-voltage-activated Ca$_v$ blockade upon activation of 1, 6, and 10 spines on ΔF/F in Sp1 (left, $n = 8$ spines in 4 cells) and dendrite (right, $n = 4$). No interaction effect of Cd$^{2+}$ wash-in after mib on spine ΔF/F (spine # × Cd$^{2+}$): F$_{(2,47)} = 1.51$, $p = 0.254$; Cd$^{2+}$ effect: F$_{(1,47)} = 14.02$, $p = 0.007$. Asterisks: significance of difference mib + Cd$^{2+}$ versus mib only. arith., arithmetic; Ca$_v$, voltage-gated Ca$^{2+}$ channel; LVA, low-voltage-activated; mib, mibefradil; O/I, output/input; uEPSP, uncaging-evoked excitatory postsynaptic potential.

cell morphology or their relative location on the dendritic tree, which indicates a highly compact dendrite and strong isolation of the spines. Clustered spines, however, facilitate Ca$^{2+}$-spike generation.

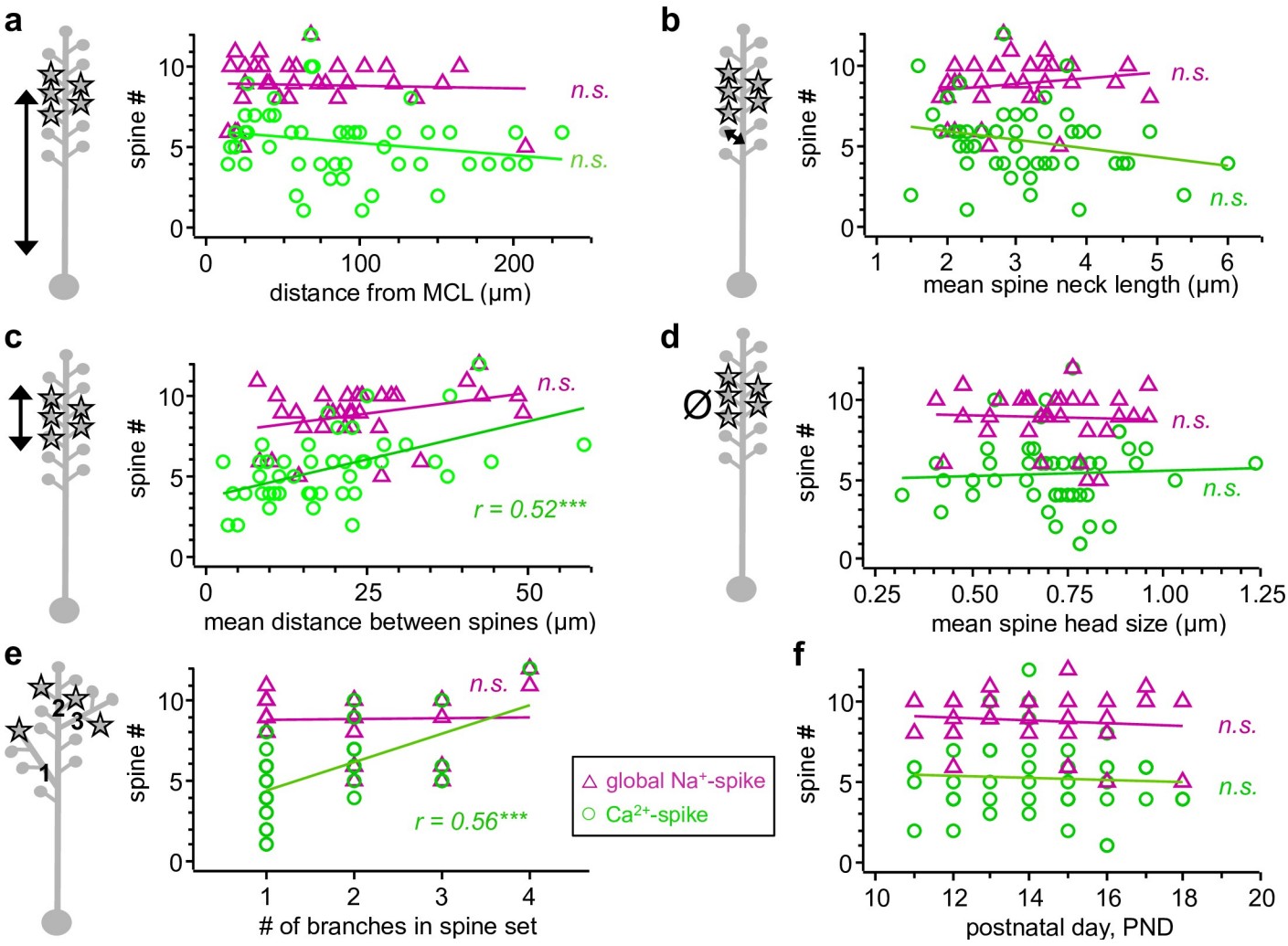

**Fig 8. Impact of morphological variables on threshold spine number for $Ca^{2+}$-spike and global $Na^{+}$-spike generation.** $Ca^{2+}$-spike data ($n = 47$ spine sets) are denoted by green circles ○ and global $Na^{+}$-spike data ($n = 31$ spine sets) by magenta triangles Δ. D-spike data not included for clarity (but see Table 1 and data repository). Linear correlation indicated by correlation coefficient r. See Table 1 for power of regressions. **a:** Influence of mean spine distance from the MCL on spine # to elicit $Ca^{2+}$-spikes ($r^2 = 0.02$, $p = 0.17$) and global $Na^{+}$-spikes ($r^2 = 0.00$, $p = 0.89$). **b:** Influence of the mean spine neck length of activated spine sets on spine # to elicit $Ca^{2+}$-spikes ($r^2 = 0.03$, $p = 0.12$) and global $Na^{+}$-spikes ($r^2 = 0.00$, $p = 0.35$). **c:** Influence of the spatial distribution of activated spines on spine # to elicit $Ca^{2+}$-spikes ($r^2 = 0.26$, $p < 0.001$) and global $Na^{+}$-spikes ($r^2 = 0.08$, $p = 0.06$). **d:** Influence of the mean spine head size (estimated diameter) that the spine set is distributed across on spine # to elicit $Ca^{2+}$-spikes ($r^2 = 0.00$, $p = 0.75$) and global $Na^{+}$-spikes ($r^2 = 0.00$, $p = 0.86$). **e:** Influence of number of different dendritic branches that the spine set is distributed across on spine # to elicit $Ca^{2+}$-spikes ($r^2 = 0.31$, $p < 0.001$) and global $Na^{+}$-spikes ($r^2 = 0.00$, $p = 0.91$). **f:** Influence of age of animal PND on spine # to elicit $Ca^{2+}$-spikes ($r^2 = 0.00$, $p = 0.83$) and global $Na^{+}$-spikes ($r^2 = 0.00$, $p = 0.58$). MCL, mitral cell layer; n.s., not significant; PND, postnatal day.

## Discussion

### High excitability of granule cell apical dendrites

Upon simultaneous multispine stimulation, granule cell dendrites can generate $Ca^{2+}$-spikes, D-spikes, and global $Na^{+}$-spikes already at rather low input numbers ($Ca^{2+}$-spike, approximately 5 inputs; D-spike, approximately 7 inputs; global $Na^{+}$-spike $\geq$ 9 inputs, Fig 9). Thus, granule cell dendrites are highly excitable. In cortical pyramidal cells, we had previously observed that global $Na^{+}$-spike generation required a similar spine number using the very same holographic system ($10 \pm 1$, $n = 7$ spine sets in 4 pyramidal cells; [25]), even though their resting $V_m$ was depolarized by $\geq$ +10 mV versus granule cells. The high granule cell excitability is not due to excessive photostimulation, because the average single EPSP amplitude was

**Table 1. Regression between coactivated threshold spine numbers for Ca²⁺-spike, D-spike, and global Na⁺-spike and various morphological variables and input patterns (see Materials and methods).**

| Parameter | Influence on generation of | | | | | | | | | | | |
|---|---|---|---|---|---|---|---|---|---|---|---|---|
| | Ca²⁺-spike (n = 47) | | | | D-spike (n = 20) | | | | global Na⁺-spike (n = 31) | | | |
| | $r^2$ | $p$ | COF | power | $r^2$ | $p$ | COF | power | $r^2$ | $p$ | COF | power |
| Spine distribution | 0.257 | <0.001 | 0.092 | 0.970 | 0.000 | 0.389 | 0.049 | 0.134 | 0.077 | 0.064 | 0.051 | 0.457 |
| Distance from MCL | 0.019 | 0.173 | −0.008 | 0.274 | 0.000 | 0.888 | −0.002 | 0.034 | 0.000 | 0.844 | −0.001 | 0.039 |
| Distance from soma | 0.022 | 0.156 | −0.007 | 0.293 | 0.000 | 0.868 | 0.002 | 0.036 | 0.000 | 0.763 | −0.002 | 0.048 |
| # of different branches | 0.310 | <0.001 | 1.713 | 0.993 | 0.000 | 0.824 | −0.178 | 0.041 | 0.000 | 0.914 | 0.035 | 0.032 |
| # of preceding bifurcations | 0.000 | 0.988 | 0.004 | 0.026 | 0.000 | 0.548 | −0.310 | 0.086 | 0.000 | 0.366 | 0.259 | 0.144 |
| Spine neck length | 0.030 | 0.120 | −0.510 | 0.343 | 0.000 | 0.424 | 0.559 | 0.121 | 0.000 | 0.355 | 0.361 | 0.149 |
| Spine head size | 0.002 | 0.748 | 0.642 | 0.050 | 0.000 | 0.849 | 0.731 | 0.038 | 0.001 | 0.862 | −0.373 | 0.037 |
| Diameter of proximal dendrite | 0.000 | 0.895 | −0.058 | 0.034 | 0.017 | 0.266 | 0.864 | 0.196 | 0.037 | 0.140 | 0.597 | 0.313 |
| Distance first branchpoint from MCL | 0.000 | 0.923 | 0.001 | 0.031 | 0.000 | 0.565 | −0.010 | 0.080 | 0.073 | 0.070 | −0.013 | 0.443 |
| Distance first branchpoint from soma | 0.000 | 0.462 | −0.005 | 0.110 | 0.000 | 0.828 | 0.003 | 0.040 | 0.046 | 0.122 | −0.009 | 0.339 |
| Single-spine uEPSP amplitude | 0.000 | 0.718 | 0.172 | 0.055 | 0.074 | 0.130 | −1.683 | 0.326 | 0.000 | 0.406 | −0.333 | 0.128 |
| Age (PND 11–18) | 0.000 | 0.833 | 0.033 | 0.040 | 0.000 | 0.578 | −0.178 | 0.077 | 0.010 | 0.577 | −0.087 | 0.080 |

Statistically significant values are highlighted in yellow. If applicable, parameter values always refer to the ensemble of spines at threshold (for Ca²⁺-spike, D-spike, and global Na⁺-spike, respectively). Thus, the value of the spine neck length for the D-spike for a given cell is the average neck length of all its spines that were activated at D-spike threshold. $p$-value is 2-tailed significance level of regression.

COF, coefficient constant; D-spike, dendritic Na⁺-spike; MCL, mitral cell layer; $n$, number of analyzed spine sets; PND, postnatal day; $r^2$, adjusted coefficient of determination; uEPSP, uncaging-evoked EPSP.

slightly smaller than in earlier reports on mitral/tufted cell to granule cell synaptic transmission [15, 24]. Thus, active dendritic mechanisms can be expected to also play a substantial role in granule cell processing in vivo, similar to what has been observed recently for cortical pyramidal cells [39, 40].

Our data also demonstrate that the full set of active dendritic mechanisms known from other neurons [1] can be triggered solely by inputs to the apical granule cell dendrite. The cells in our sample were located close to the mitral cell layer and thus belong to superficial granule cells [14], which are reportedly more excitable than deep granule cells [21]. Thus, our results might not generalize to all granule cell subtypes, possibly explaining the discrepancy with earlier excitability estimates (see Introduction, [24]).

The global Na⁺-spike threshold spine number reported here is a lower limit, since in approximately two thirds of cells in our sample full-blown somatic Na⁺-spikes could not yet be elicited at the maximal number of 10–12 activatable inputs (see Materials and methods). Morphological variables did not influence Na⁺-spike thresholds, indicating that the superficial granule cell's dendritic tree is electrotonically compact.

The low-threshold spine number seems to match previous observations that uniglomerular stimulation can already fire granule cells [22, 41, 42]. Because in total approximately 20 mitral and tufted cells are estimated to belong to a glomerular column, with a slightly lower share of tufted cells [43–45], and the release probability at these inputs is approximately 0.5 [17], a given granule cell is unlikely to be fired solely from intracolumnar dendrodendritic inputs, requiring additional activation perhaps via mitral/tufted cell axonal collaterals [46]. However, uniglomerular inputs—if clustered—might suffice to elicit local Ca²⁺-spikes, and mitral/tufted cell theta bursts as observed in vivo [47] could also trigger firing of intracolumnar granule cells from the distal apical dendrite.

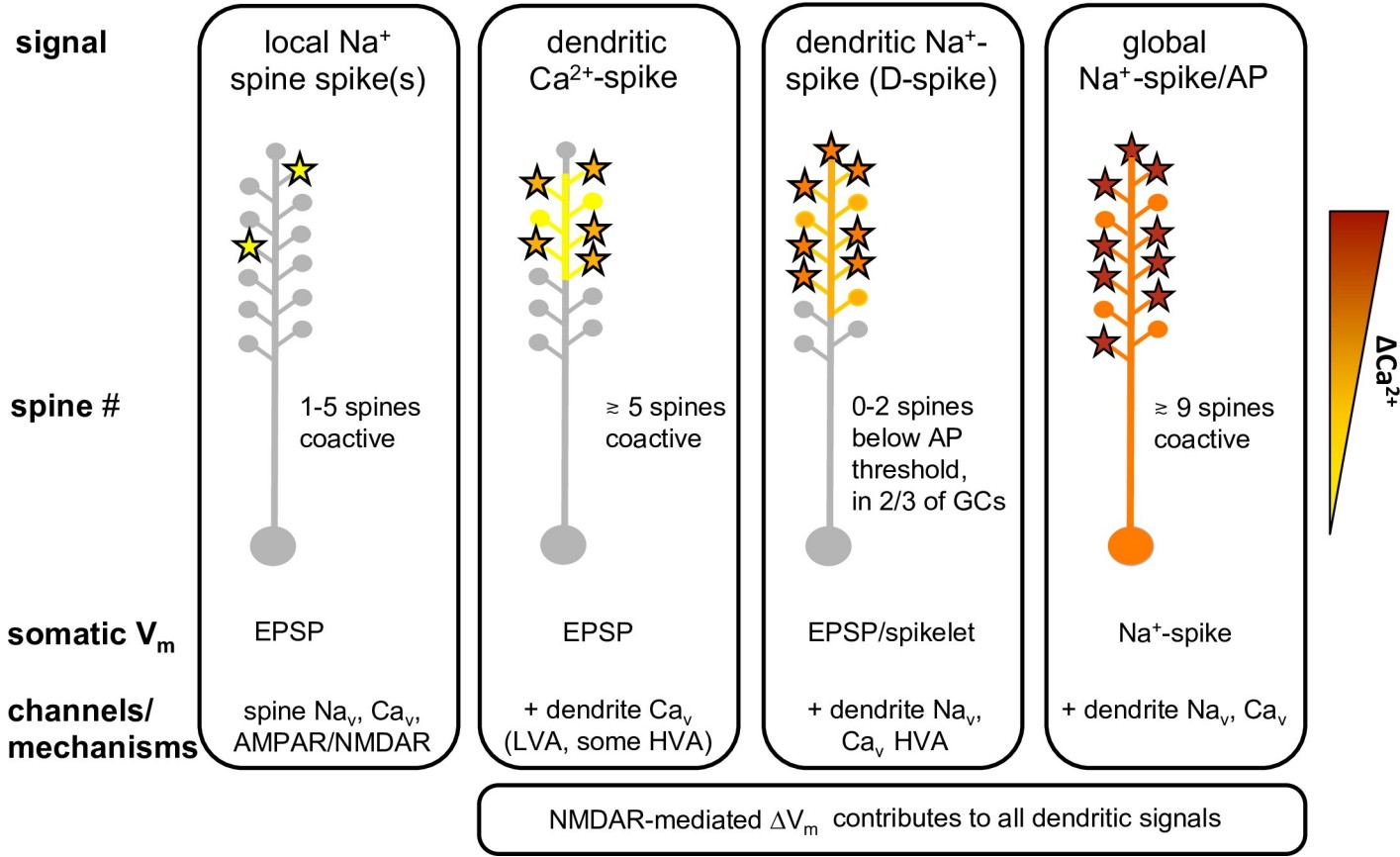

**Fig 9. Summary of findings: Active signal types in granule cell dendrites upon simultaneous stimulation of apical dendritic spines.** Graphical summary of main results. Left: Spine #: number of coactivated spines. Somatic $V_m$: somatic membrane potential response; no shape changes observed here for regional dendritic $Ca^{2+}$ spikes. Channels/mechanisms: components involved in generating the respective $Ca^{2+}$ and $V_m$ signals, located in the excited spines and, for all nonlocal signals, excited dendritic segments. Right: Color scale for $\Delta Ca^{2+}$ entry. AMPAR, AMPA receptor; AP, action potential; D-spike, dendritic $Na^+$-spike; GC, granule cell; HVA, high voltage activated; LVA, low voltage activated; NMDAR, NMDA receptor.

## Dendritic spiking: D-spike and localized $Ca^{2+}$-spike

In the majority of granule cells, we detected D-spikes correlated with the onset of supralinear integration at the soma, either as distinct spikelets, or if these were masked by electrotonic filtering, by step-like increases in the compound uEPSP rate of rise, rise time, decay, and spine $\Delta F/F$ (Fig 3). $Na^+$-spikelets were not reported from juvenile rat granule cells before and probably emerged here because of clustered stimulation. Increases in the EPSP rate of rise and dendritic $\Delta Ca^{2+}$ also indicated D-spikes in CA1 pyramidal cells and mouse and frog granule cells [3, 20, 21]. The unexpected increase in granule cell compound uEPSP rise time by almost approximately 10 milliseconds observed here can be explained by the substantial latency of spikelets of approximately 10–20 milliseconds after stimulation onset. This delay indicates that D-spikes are not spatially expanded spine spikes, implying that spine spikes will not invade the dendrite even under conditions of clustered spine activation. This notion is further supported by the lack of a correlation between spatial clustering and D-spike or global $Na^+$-spike threshold spine numbers.

Rather, single EPSPs are strongly attenuated and also temporally filtered across the spine neck because of its high resistance [15], resulting in slowed integration. A-type $K^+$ currents are known to delay granule cell firing [48] and thus, may also contribute to the delay of D-spikes

and the yet longer latency of global Na$^+$-spikes at threshold (approximately 40 milliseconds). Initiation of D-spikes most likely happens at dendritic Na$_v$ hot-spots [49], whereas the existence of a dedicated global Na$^+$-spike initiation zone in granule cell apical dendrites seems probable, with its precise location a matter of speculation at this point (but see [18]).

All granule cells in our sample featured Ca$^{2+}$-spikes (in terms of dendritic Ca$^{2+}$ entry), which, at threshold, were regional and did not influence somatic V$_m$. Although Ca$_v$ densities are apparently lower in the proximal apical dendrite [33], the Ca$^{2+}$-spikes evoked by glomerular stimulation in our previous study [17] occurred in an all-or-none fashion throughout the entire dendritic tree with a concomitant increase and broadening of somatic EPSPs that were not observed here. The distribution of glomerular inputs across the granule cell dendrite is not yet known but might well be more dispersed than the maximal accessible extent in this study and rather likely also involves mitral cell axonal inputs to the basal dendrites [46, 50, 51]. Somatic depolarization of granule cells can generate the classical T-type Ca$_v$-mediated humps in V$_m$ [33, 52]. Therefore, the main initiation zone for such global Ca$^{2+}$-spikes evoked by glomerular stimulation is probably not located in the distal apical dendritic tree (see also [18]). Thus, if input to densely packed spines can cause regional Ca$^{2+}$-spikes, these might provide a substrate for local lateral inhibition, as suggested earlier [35, 53, 54]. In any case, local Ca$^{2+}$-spikes became more global close to the global Na$^+$-spike threshold, along with recruitment of high-voltage-activated Ca$_v$s. Thus, granule cell dendrites feature multiple levels of compartmentalization.

In contrast to global Na$^+$-spike generation, Ca$^{2+}$-spike generation was strongly influenced by input distribution, in line with electrotonic attenuation of subthreshold EPSPs along the dendrite [1]. Because the Ca$^{2+}$-spike precedes the D-spike and global Na$^+$-spike and its space constant of at least 60 μm covers the maximum spatial extent of spine sets in our experiments, its presence can reduce passive attenuation and thus, explain the observed independence of D-spike and global Na$^+$-spike generation from spatial input distributions (within the accessible spatial regime investigated here).

## NMDA-spikes and role of NMDARs in granule cell synaptic processing

NMDARs contribute substantially to supralinear integration in granule cells, both at the level of V$_m$ and ΔCa$^{2+}$. They are required for Ca$^{2+}$-spike generation, and their blockade had a much stronger effect on V$_m$ supralinearity than Na$_v$ or Ca$_v$ blockade. This higher efficiency is probably related to the slower kinetics of the NMDAR current, which is filtered much less both by the spine neck and along the dendritic tree compared with Na$_v$ or Ca$_v$-mediated currents [55]. The substantial impact of NMDARs on granule cell dendritic integration is characteristic for NMDA-spikes [56, 57]. Accordingly, granule cell global Na$^+$-spikes evoked by synaptic stimulation are followed by NMDAR-dependent plateau potentials [22, 23]. In most cells investigated here, compound uEPSP half durations extended >50 milliseconds for higher spine numbers, thus dendritic Ca$^{2+}$- and Na$^+$-spikes are closely intertwined with NMDA-spikes.

As a note of caution, holographic uncaging might overemphasize the role of NMDARs, because (1) APV blocks uncaging-evoked spine ΔF/F slightly more than synaptic ΔF/F (to 65% versus 50% of control; [15]) and (2) the axial point spread function of our multisite uncaging system is extended to 2.7 μm from 1.1 μm [25], possibly covering yet more extrasynaptic NMDARs. However, the effect on APV on single-spine uncaging-evoked ΔF/F was similar as in Bywalez and colleagues [15].

NMDARs are predicted to enable supralinear summation of ΔCa$^{2+}$ at positive Hebbian pairing intervals of single-spine spike and global Na$^+$-spikes [58]. The global Na$^+$-spike latency at threshold of 40 milliseconds observed here matches the simulated regime of maximally

supralinear summation efficiency, which explains the strong increase of $\Delta Ca^{2+}$ in spines upon global $Na^+$-spike generation. In conclusion, NMDARs are essentially involved in all aspects of granule cell reciprocal synaptic processing, including release of GABA from reciprocal spines [16] and synaptic plasticity [59, 60].

### Functional implications for olfactory processing

Dendritic spikes and therewith possibly lateral inhibition can be invoked already at very low numbers of coactivated granule cell spines. The stepwise increases in spine and dendrite $\Delta Ca^{2+}$ at the 3 spike thresholds observed here and in earlier work [17, 22] imply that the fairly low release probability for GABA ($P_{r\_GABA}$, approximately 0.3 for local stimulation [16]) might also be increased in a stepwise fashion via the summation of local spine spikes and nonlocal spike types. Such coincident activation could render both lateral and recurrent inhibition more effective and is likely to represent the standard scenario for granule cell–mediated lateral inhibition, according to our recent hypothesis [16].

In any case, granule cell–mediated lateral inhibition is thought to implement contrast enhancement and synchronization of gamma oscillations across glomerular columns responding to the same odorant [61–64]. Fast gamma oscillations in the bulb are generated at the reciprocal synapse between granule cells and mitral/tufted cells, independently of global $Na^+$-spikes [65, 66], and require a fast excitatory–inhibitory feedback loop [62, 67] that is likely to involve both reciprocal and lateral processing. D-spikes could be powering such fast oscillatory lateral and recurrent output because of their shorter latencies versus global $Na^+$-spikes. Zelles and colleagues [20] already proposed an interaction of D-spikes and back-propagating global $Na^+$-spikes in granule cells at intervals as short as 5 milliseconds, and also Pinato and Midtgaard [19] could elicit spikelets at a frequency of 150–250 Hz, whereas the maximum frequency of global $Na^+$-spikes was much lower (10–30 Hz). In vivo, granule cell global $Na^+$-spike firing was found to be sparse under anesthesia [68, 69] and increased in awake animals [69, 70], albeit not up to the gamma range, whereas spikelets have been frequently observed [71–74]. Similarly, D-spikes are associated with sharp wave-associated ripples (120–200 Hz) in hippocampal CA1 pyramidal cells [75–77]. Although granule cells are unlikely to drive slow bulbar theta oscillations [62], excitatory inputs to granule cells can be coupled to the respiratory rhythm with variable phases [78]. Therefore, similar to the theta bursts observed in mitral and tufted cells, the firing of granule cell D-spikes might also occur in a spaced fashion. At the level of the local field potential, the respiratory phase could couple to the amplitude of these fast events (as illustrated in Fukunaga and colleagues 2014 [62], their supplementary Fig 1), which, in turn, might be modulated by olfactory learning and allow to encode context at the level of the bulb [79].

According to our observations, granule cell spine and dendrite $Ca^{2+}$ entry were not necessarily correlated with changes in somatic $V_m$ amplitude (both for the localized $Ca^{2+}$-spike and the attenuated D-spike), allowing for multiplexed signals, as proposed for cerebellar granule cells [80], which, in bulbar granule cells, might implement, e.g., independent plasticity induction across reciprocal spines [59]. On a yet more speculative note, different granule cell spike types might encode different aspects of odor information. Such multiplexing of odor information was already described for mitral cells in zebrafish, in which gamma oscillations and tightly phase-locked spiking were observed to be tied to odor category and odor identity, respectively [81].

## Materials and methods

### Ethics statement, animal handling, slice preparation, and electrophysiology

All experimental procedures were performed in accordance with the rules laid down by the EC Council Directive (86/89/ECC) and German animal welfare legislation. According to this

legislation (§4 Absatz 3 TierSchG), the preparation of acute brain slices for in vitro experiments by certified personnel (which applies to both MM and VE) is monitored by the institutional veterinarian of Regensburg University and does not require approval by an ethics committee. Rats (postnatal day 11–21, Wistar of either sex) were deeply anaesthetized with isoflurane and decapitated. Horizontal olfactory bulb brain slices (thickness 300 μm) were prepared and incubated at 33˚C for 30 minutes in ACSF bubbled with carbogen and containing (in mM): 125 NaCl, 26 NaHCO$_3$, 1.25 NaH$_2$PO$_4$, 20 glucose, 2.5 KCl, 1 MgCl$_2$, and 2 CaCl$_2$. Recordings were performed at room temperature (22˚C). Patch pipettes (pipette resistance 5–7 MΩ) were filled with an intracellular solution containing (in mM): 130 K-methylsulfate, 10 HEPES, 4 MgCl$_2$, 2.5 Na$_2$ATP, 0.4 NaGTP, 10 Na-phosphocreatine, 2 ascorbate, 0.1 OGB-1 (Ca$^{2+}$ indicator, Invitrogen), 0.04–0.06 Alexa Fluor 594 (Life Technologies) (pH 7.3). The following pharmacological agents were added to the bath in some experiments: TTX (0.5–1 μM, Alomone), D-APV (25 μM, Tocris), mibefradil (10 μM, Tocris), and cadmium chloride (Cd$^{2+}$, 100 μM, Sigma). After control recordings, drugs were washed in for at least 10 minutes before restarting recordings. Electrophysiological recordings were made with an EPC-10 amplifier and Patchmaster v2.60 software (both HEKA Elektronik). Granule cells were patched in whole-cell current clamp mode and held near their resting potential of close to –75 mV [33]. If granule cells required >25 pA of holding current, they were rejected. In order to provide optimal optical access to the granule cell apical dendritic tree, patched cells were located close to the mitral cell layer (mean depth 14 ± 12 μm, $n$ = 63 cells).

## Combined 2-photon imaging and multisite uncaging in 3D

Imaging and uncaging were performed on a Femto-2D-uncage microscope (Femtonics). The microscope was equipped with a 60× water-immersion objective used for patching (NA 1.0 W, NIR Apo, Nikon) and a 20× water-immersion objective used for 2-photon imaging and uncaging (NA 1.0, WPlan-Apo, Zeiss). Green fluorescence was collected in epifluorescence mode. The microscope was controlled by MES v4.5.613 software (Femtonics). Two tunable, verdipumped Ti:Sa lasers (Chameleon Ultra I and II, respectively, Coherent) were used in parallel, set to 835 nm for excitation of OGB-1 and to 750 nm for uncaging of 4-methoxy-5,7-dinitroindolinyl-caged glutamate (DNI, Femtonics; [82]). DNI was used at 0.6 mM concentration in a closed perfusion circuit with a total volume of 12 ml and was washed in for at least 10 minutes before uncaging. To visualize the spines and for Ca$^{2+}$ imaging, we waited at least 20 minutes for the dyes to diffuse into the dendrite before imaging.

Imaging and uncaging laser beams were decoupled before the entrance of the galvanometer-based 2D scanning microscope to relay the uncaging beam to a spatial light modulator (SLM X10468-03, Hamamatsu). Next, we positioned the multiple uncaging spots/foci in 3D at a distance of 0.5 μm from the spine heads, using custom-written software (based on Matlab). The holographic projector module and software are described in detail in Go and colleagues [25]. The available laser power at the sample of our system allowed for a maximum number of 12 spots in a volume of 70×70×70 μm$^3$. Usually spines no deeper than approximately 30 μm were imaged, because otherwise uncaging laser power was too much attenuated. The positioning was checked before each measurement and, if necessary, readjusted to account for drift. The uncaging pulse duration was 1–2 milliseconds, and the laser pulse power was adjusted individually for each experiment to elicit physiological responses [15]. For simultaneous multisite photostimulation, the total uncaging power and the number of uncaging spots were kept constant. "Superfluous" foci, i.e., foci that were not needed as stimulation spots at a given time of an experiment, were excluded by positioning them just outside the holographic field-of-view, such that they would fall off the optics and not be projected onto the sample [25].

Imaging of uncaging-evoked $Ca^{2+}$ signals in selected spines and dendritic positions within one 2D plane was carried out as described earlier [15]. During simultaneous $Ca^{2+}$ imaging and photostimulation, imaging was started 700 milliseconds before the uncaging stimulus. During uncaging, the scanning mirrors were fixed.

In each experiment, single spines were consecutively activated and somatic single-spine uncaging EPSPs (uEPSPs) were recorded for each spine separately. Next, successively increasing numbers of these spines were simultaneously activated, and somatic compound uEPSPs were recorded until the cell fired an action potential (AP, global $Na^+$-spike) or, in the experiments with focus on subthreshold integration, until a maximum number of 10 activated spines was reached. A subset of spines and dendritic locations located within the same focal plane were chosen for 2-photon line-scanning to gather $Ca^{2+}$ imaging data. At least one spine, termed spine 1 in the following, was always located in this imaging plane to gather complete $\Delta F/F$ data sets from the activation of only this single spine to the additional activation of more and more spines until the maximal number. Because of the spine density being higher in distal regions and $Ca^{2+}$ imaging being restricted to 1 focal plane, most dendritic measurements at a distance from the center of the stimulated spine set (Fig 1B) were still proximal to the stimulation site. The sequence of the additional successively activated spines with respect to their position on the dendritic tree was chosen randomly. However, the low spine density (see Introduction) and the restriction to a volume of $70 \times 70 \times$ approximately $30 \ \mu m^3$ mostly determined the choice of activated spines. Both single-spine stimulations and the different combinations of multisite uncaging were, if possible, performed at least twice, and recordings were averaged for analysis.

Because such experiments were performed with up to 40 different stimulation conditions, we decided to increase the spine numbers by increments of +2 for some experiments (in particular, for pharmacology) in order to limit the experiment duration and thus to ensure a good recording quality.

## Data analysis

Changes in $Ca^{2+}$ indicator fluorescence were measured relative to the resting fluorescence $F_0$ in terms of $\Delta F/F$, as described previously [17]. Electrophysiological and $Ca^{2+}$ imaging data were analyzed using custom macros written in IGOR Pro (Wavemetrics). As in our previous studies, spontaneous activity was high in general, and traces contaminated by such activity during baseline just before uncaging or during the rising phase of the uEPSP were discarded. Multiple (2 or more) recordings of the same stimulation type were averaged and smoothed (box smoothing) for analysis. uEPSP and $\Delta F/F$ rise times were analyzed in terms of the interval between 20% and 80% of total uEPSP/$\Delta F/F$ amplitude; uEPSPs and $\Delta F/F$ half durations ($\tau\_1/2$) were analyzed in terms of the interval between the peak and 50% of the total EPSP or $\Delta F/F$ amplitude. The uEPSP maximum rate of rise was determined by the peak of the first derivative of the uEPSP rising phase. The global $Na^+$-spike threshold was detected via the zero point of the second derivative of the rising phase of the action potential.

Integration was quantified by plotting the amplitude of the arithmetic sum of the respective single uEPSP traces versus the actually measured multispine compound uEPSP amplitude for increasing numbers of coactivated spines, yielding an sO/I (from [1], where these plots are termed sI/O). If the compound uEPSP amplitude consistently exceeeds the amplitude of the arithmetic sum of the single uEPSP traces beyond a certain stimulation strength by at least a factor of 1.2, we classified these sO/I patterns as supralinear. If the factor fell consistently below 0.8, we classified these sO/I patterns as sublinear, and the patterns falling between these categories were considered to be linear. The factors were set at 0.8 and 1.2 to exceed potential

undersampling errors in uEPSP amplitudes (see next section). The supralinearity criterion was empirically confirmed by concurrent characteristic changes in compound uEPSP kinetics (increase in rate of rise due to the D-spike, see Fig 3) and further validated by variation (see S1 Table).

As criterion for the presence of a $Ca^{2+}$-spike, dendritic $Ca^{2+}$ transient amplitudes $\Delta F/F$ had to exceed a value well above noise level ($\geq$ 8% $\Delta F/F$ or factor 1.5 above noise level of 5% $\Delta F/F$) and be detectable at every dendritic line scan located within the section of the dendrite carrying stimulated spines (Fig 2, Fig 4E; [17]).

## Data sampling, normalization, and alignment

Because for any particular number of coactivated spines we could usually perform no more than 2 stimulations in the interest of finishing experiments within the average life time of granule cell recordings, the individual compound uEPSP measurements might differ from the mean for that particular spine number due to undersampling. For single uEPSPs, a previous data set of spines with higher numbers of samplings (from [15], obtained on the same experimental rig except for the added spatial light modulator) allowed to estimate the variance at the average uEPSP amplitude of 1.40 mV in the experiments in this study as 0.39 mV ($n$ = 18 spines, S1B Fig) and also to determine the number of samplings required to properly detect the variance between uEPSP measurements from a given spine ($n$ = 6). Thus, a sampling number of 2 per uEPSP as in the current data set will increase the general variance by a factor of approximately $\sqrt{(6/2)} = \sqrt{3}$ [83]. On the other hand, stimulations of larger numbers of spines $N_{spine}$ will reduce this sampling problem by a factor of $\sqrt{N_{spine}}$, similar to the effect of averaging across repeated stimulations of the same spine [83]; the same argument holds for the arithmetic summation of the involved $N_{spine}$ single uEPSPs. S1C Fig shows the resulting prediction of the variances in the EPSP amplitude for linear summation, assuming that all single uEPSPs are of similar size, because a difference in size should not affect linearity. Based on this estimate, we expect to be able to detect deviations from linear behavior by more than ±0.2 beyond approximately 5 costimulated spines.

Similarly, the variance in single-spine $\Delta F/F$ is on the order of 6% $\Delta F/F$ or approximately 0.2 of the total signal (again derived from [15]). To compare spine 1 $\Delta F/F$ amplitude multispine activation data across experiments relative to solely local activation of spine 1, we intended to normalize these to the spine 1 $\Delta F/F$ amplitude for unitary activation. Because of the undersampling problem, we tested for up to which spine number there was no significant increase in $\Delta F/F$, which yielded 4 spines (Friedman repeated measures ANOVA on ranks: $X^2_F(3) = 4.80$, $p$ = 0.187). Therefore, we averaged spine 1 $\Delta F/F$ for (co)stimulations of 1, 2, 3, and 4 spines and used the mean as basal unitary $\Delta F/F$ for normalization. Thus, undersampling of spine $\Delta F/F$ could be compensated for by this means.

Because dendritic $\Delta F/F$ was usually not detectable for low numbers of stimulated spines, normalization to the average dendritic $\Delta F/F$ in response to stimulation of spines 1, 2, 3, and 4 would have introduced a very high variance. Instead, $\Delta F/F$ in the dendrite was normalized to the mean of all responses from the onset of the dendritic $Ca^{2+}$-spike until below global $Na^+$-spike threshold.

Because each granule cell required its individual spine number to reach the thresholds for the nonlocal events $Ca^{2+}$-spike, D-spike, and global $Na^+$-spike (for the respective pattern of stimulation), we aligned the data in relation to the onset of the nonlocal event (e.g., Fig 2D and 2E relative to $Ca^{2+}$-spike). Such realignments allow us to reveal effects across the sampled cells that otherwise would be smeared out because of cell-specific thresholds, such as recruitment of active conductances near thresholds [3].

## Morphological analysis

Granule cell apical dendrites were reconstructed from 2-photon fluorescence z-stacks gathered at the end of each experiment, using Neurolucida (MBF Bioscience). Distances were measured along the dendrite. Mean distances of a spine set from the soma or the mitral cell layer were analyzed in terms of the average distance of all stimulated spines from the soma or crossing of the apical dendrite into the mitral cell layer, respectively. The distribution of a stimulated spine set across the dendrite was analyzed in terms of the mean distance of each spine from all other stimulated spines along the dendrite. Because the degree of z-resolution in our 2-photon stacks did not allow for proper deconvolution, spine neck lengths and spine head sizes were estimated as described before [15, 84].

## Statistics

Statistical tests were performed in Sigmaplot 13.0 (Systat Software, Inc) or on vassarstats.net. To assess statistical significance levels across spine numbers or threshold $V_m$ values for $Ca^{2+}$-spike versus global $Na^+$-spike (Fig 2C), data sets were compared using paired $t$-tests for dependent data sets. Not normally distributed data sets (Shapiro-Wilk Normality Test) were compared using Wilcoxon signed rank tests. To assess statistically significant differences from linear summation in sO/I relation data sets, the distribution of ratios of the measured uEPSP amplitudes/arithmetic sums (O/I ratio) was tested against a hypothesized population mean/median of 1.0 (corresponding to linear summation), using 1-sample $t$-tests or 1-sample signed rank tests for not normally distributed data. To assess variation in repeated measure data sets (Fig 3C) repeated measures ANOVA together with all pairwise multiple comparison procedure (Holm–Sidak method) was performed. For pharmacology experiments (e.g., Fig 5D) repeated measures 2-way ANOVA together with all pairwise multiple comparison procedure (Holm–Sidak method) was performed. For statistical analysis of dendritic ΔF/F before and after pharmacological treatment, just stimulations of ≥4 spines were taken into account, because for lower numbers of spines, usually no signal was detectable under control conditions.

Because of the increase of spine numbers by increments of 2 in some experiments, averaged data points for a given spine number across experiments do not contain the same $n$ of individual measurements as for other spine numbers. Even more so when the data were aligned relative to individual spike thresholds (e.g., alignment relative to $Ca^{2+}$-spike threshold in Fig 2D and 2E), because not all experiments contained data points for the more remote spine numbers [+2] or [−3] relative to threshold. In addition, in experiments with data gaps just before a global spike threshold at spine number [x], it is not possible to know whether the spike threshold could have already been reached at [x-1] spines (e.g., alignment relative to $Ca^{2+}$-spike in Fig 2D and 2E). We accounted for this uncertainty by averaging the data in the continuous experiments for [−2] and [−1] and used these averaged data together with the data from experiments with gaps for paired comparison of parameters below and at threshold (nonparametric Wilcoxon test). S2 Fig shows the individual data points for all these comparisons normalized to [−2/−1]. If there was a significant linear increase or decrease with spine number in the parameter in the subthreshold regime (grey dashed lines in Fig 2D and 2E; Fig 3D and 3E; Fig 4C–4F), the expected increment based on this change was subtracted from the parameter values at threshold before statistical testing for a difference.

To assess statistical significance for linear increase and decrease (Table 1) we performed a linear regression analysis. Given $r^2$ values are adjusted $r^2$ values.

## Supporting information

**S1 Fig. Distribution of single uEPSP amplitudes, variance of single uEPSPs and estimate of variance for compound and summated uEPSPs. a**: Blue histogram: amplitude distribution

of single-spine uEPSPs in this study ($n$ = 272 spines). Mean uEPSP: 1.4 ± 1.4 mV. Gray histogram: distribution of uEPSPs from previous study (Bywalez and colleagues 2015; right axis, $n$ = 47 spines). **b**: Recordings of multiple uEPSP responses from the same spine ($n$ = 9 ± 3 responses on average, $n$ = 18 spines, mean uEPSP 1.6 ± 1.0 mV) from Bywalez and colleagues 2015, analyzed for their SD. Highly significant correlation ($p$ < 0.001), linear fit shown. For the mean value of uEPSPs in the current study of 1.4 mV, $SD_{single-spine}$ is thus on the order of 0.4 mV (blue arrows). **c:** Extrapolation of multispine EPSP amplitudes versus the arithmetic single-spine EPSP sum for linear summation from b and the same mean single uEPSP and $SD_{single-spine}$ response for all spines (as extrapolated from b). Variations of mean EPSP size across spines were not taken into account because these should not influence the linearity of summation. White numerals: respective spine number. Error bars in the x-dimension (arithmetic sum): Black: Expected standard deviation $SD_{sum}$ for ideal recording conditions (at least 6 stimulations per spine, $SD_{sum} = (\sqrt{N_{spines}})^* SD_{single-spine}$, see Methods). Blue: standard deviation of EPSP amplitudes in our data set extrapolated from the ideal SD. Because there are only 2 stimulations per spine instead of the 6 stimulations required to properly measure $SD_{single-spine}$, the actual $SD_{single-spine}$ is increased by a factor of $\sqrt{3}$ compared with the ideal $SD_{single-spine}$ and thus the $SD_{sum}$ is also increased by a factor of $\sqrt{3}$ (see Methods). Error bars in the y-dimension (compound EPSP): Black: Expected standard deviation $SD_{multi-spine}$ for ideal recording conditions and linear summation of similar uEPSPs (at least 6 stimulations per spine set, $SD_{multi-spine} = (\sqrt{N_{spines}})^* SD_{single-spine}$, see Methods). Blue: SD of EPSP amplitudes in our data set extrapolated from the ideal SD, similar to the x-dimension: since there are only 2 stimulations per spine set, the actual $SD_{multi-spine}$ is increased by a factor of $\sqrt{3}$ compared to the ideal $SD_{multi-spine}$. uEPSP, uncaging-evoked excitatory postsynaptic potential.
(DOCX)

**S2 Fig. Individual data sets at threshold for Ca$^{2+}$-spike and D-spike.** Individual data points from paired data comparisons across threshold for Ca$^{2+}$-spikes (**a**), D-spikes (**b**) and effect of TTX on D-spike transitions (**c**). These data were not plotted in the main figures for sake of clarity. In **a**, **b** data are shown normalized to the average value below threshold (except for ΔF/F dendrite because of several points with value zero) and corrected for linear trend in subthreshold data (see Methods). In **c**, changes Δ in parameter values across threshold in TTX are shown normalized to their increase Δ in control, thus no correction for linear trends is required. Analysis of half duration is missing because there were not enough data points for statistical analysis. $^*p$ < 0.05, $^{**}p$ < 0.01, $^{***}p$ < 0.001. D-spike, dendritic Na$^+$-spikes; TTX, tetrodotoxin.
(DOCX)

**S1 Table. Robustness of supralinearity criterion O/I ratio ≥ 1.2.** The criterion was varied by ± 0.1 and the respective data of the individual cells were rearranged accordingly before averaging. O/I, output/input.
(DOCX)

## Acknowledgments

We thank Anne Pietryga-Krieger for expert technical assistance, and Sara Aghvami and lab members for discussions. We thank Mary Ann Go, Michael Lawrence Castanares, and Vincent R. Daria for ongoing advice with regard to holographic uncaging.

## Author Contributions

**Conceptualization:** Max Mueller, Veronica Egger.

**Data curation:** Max Mueller, Veronica Egger.

**Formal analysis:** Max Mueller, Veronica Egger.

**Funding acquisition:** Veronica Egger.

**Investigation:** Max Mueller.

**Methodology:** Max Mueller, Veronica Egger.

**Project administration:** Veronica Egger.

**Software:** Veronica Egger.

**Supervision:** Veronica Egger.

**Validation:** Veronica Egger.

**Visualization:** Max Mueller, Veronica Egger.

**Writing – original draft:** Max Mueller, Veronica Egger.

**Writing – review & editing:** Max Mueller, Veronica Egger.

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
