## [Editor Report · Decision Letter 0]

2 Apr 2020

Dear Dr Egger, 

Thank you for submitting your manuscript entitled "Dendritic integration in olfactory bulb granule cells upon simultaneous multi-spine activation: Low thresholds for non-local spiking activity" for consideration as a Research Article by PLOS Biology.

Your manuscript has now been evaluated by the PLOS Biology editorial staff, as well as by an Academic Editor with relevant expertise, and I am writing to let you know that we would like to send your submission out for external peer review.

Please re-submit your manuscript within two working days, i.e. by Apr 06 2020 11:59PM.

Kind regards,

Gabriel Gasque, Ph.D.,

Senior Editor

PLOS Biology

---

## [Decision Letter · Decision Letter 1]

6 May 2020

Dear Dr Egger,

Thank you very much for submitting your manuscript "Dendritic integration in olfactory bulb granule cells upon simultaneous multi-spine activation: Low thresholds for non-local spiking activity" for consideration as a Research Article at PLOS Biology. Your manuscript has been evaluated by the PLOS Biology editors, by an Academic Editor with relevant expertise, and by three independent reviewers. You will note that reviewer 1, Diego Restrepo, has signed his comments.

In light of the reviews (below), we are pleased to offer you the opportunity to address the comments from the reviewers in a revised version that we anticipate should not take you very long. We will then assess your revised manuscript and your response to the reviewers' comments and may consult the reviewers again.

We expect to receive your revised manuscript within 1 month.

**IMPORTANT - SUBMITTING YOUR REVISION**

Your revisions should address the specific points made by each reviewer. As you will see, all reviewers, despite the general enthusiasm, think you should carefully re-write your manuscript to make it more accesible to the broad readership of PLOS Biology. Having discussed the reviewers' comments with the Academic Editor, we think you should really pay attention to this general concern. In addition, you should thoroughly address, likely by showing more data/analyses, reviewer 1's major point 2.

Please submit the following files along with your revised manuscript:

*Resubmission Checklist*

*Published Peer Review*

*PLOS Data Policy*

*Blot and Gel Data Policy*

Sincerely,

Gabriel Gasque, Ph.D., 

Senior Editor

PLOS Biology

REVIEWS:

Reviewer #1, Diego Restrepo: The manuscript by Muller and Egger presents a study of the non-linear integration of multi spine activation in granule cells of the olfactory bulb. The authors use a novel holographic stimulation method to compare activation at will of subsets of spines separately or simultaneously. This is a novel approach that yields interesting data that is potentially of interest, not only to those understanding olfactory function, but to the broader readership interested in mechanisms of non-linear integration in dendrites relevant to neural function processing in areas such as cortex, hippocampus and cerebellum. My enthusiasm is diminished significantly by the deficient presentation of the data and the lack of validation of the parameters used to classify supralinear changes in the single spine EPSP sum (suEPSP) vs. the compound multiple spine EPSP (cuEPSP).

Major criticisms

1. The presentation of the data is not clear and is currently not suitable for the broad readership of PLoS Biology. In particular, the data are presented in a manner that is difficult to follow. For example, presentation of the section on transition to supralinear behavior with D-spikes (starting in line 212) is confusing. First, the authors start by discussing Fig. 3c, without an introduction to the D-spine (Figs. 3a and b are not mentioned in the text). Second, the wording is confusing (e.g. please re-word the sentence starting with "Since Ca2+-spike…" in line 213). Third, it does not help that in most panels there is not a label of what the different colors/symbols mean (e.g. what is green vs black in Fig. 3b?). I find these problems throughout the results. Please revise the flow of the text in the results and presentation of all the figures. Please refer to all panels in the text and cite the panels in alphabetical order (if possible).

2. The criterion for classification of supralinearity at 0.2 is arbitrary. The authors provide a justification in the methods, but I am not convinced this is a well-justified choice. Please present data indicating why this criterion was chosen. Is there a statistical approach that could justify the choice (e.g. bootstrap prediction of the cuEPSP at the higher number of spines using the results for the first few spines?). Or, what happens when the arbitrary criterion is varied? Incidentally, please provide a more comprehensive introduction of quantification of integration and the supralinear criterion in the results. 

3. The authors are well-versed in dendritic processing, and I enjoyed reading the discussion. However, the section on functional implications needs reworking. First, the text indicates that in vivo GCs firing is sparse (line 639). This is correct in anesthetized animals, but it is very different in the awake animal (Cazakoff et al. 2014). Second, there needs to be a better discussion of what this would mean in terms of phase amplitude coupling in awake animals (see Youngstorm and Strowbridge 2015, Zhouang et al 2019) that has been recently implicated in conveying contextual identity (Losacco et. al. 2020).

Minor issues

1. For a reader unfamiliar with the different modes of excitability it is difficult to follow the denomination of the spikes. Please revise the definition of the different modes (e.g. D-spikes) in the introduction (line 41) to make it absolutely clear what these are, and use the same nomenclature throughout. Maybe an introductory figure would help?

2. Lines 33-36. Please provide a citation for the sentence claiming within glomerular column communication of M/Ts (do the authors mean gap junctions and/or juxtaglomerular inhibitory interneuron connections?).

3. Lines 87-89. Please show histograms for the number of APs with maximal stimulation and for the average suEPSP.

4. The authors should always provide the number of observations when they present a statistical estimate, or test. Please revise throughout (e.g. in line 123 the average, and SD? Is presented for the number of spines for supralinearity without stating the n). Please give the n whenever a statistical test/value is presented.

5. Please give the statistics for the statement in lines 268-270.

6. Line 441. Please state the specificity of mibefradil (L vs T type) at the concentration used.

7. Figs. 2d,e. The horizontal bars denoting statistical significance are misplaced? Also, was there a correction for multiple comparisons?

8. For Fig. 3d, right panel there are no asterisks. Are the differences not significant? Also see Fig. 4e.

9. Please provide error bars for Fig. 4b.

Reviewer #2: In this manuscript, Mueller and Egger describe the physiological signatures of three types of nonlinear events observed in granule cells of the olfactory bulb. Using simultaneous uncaging afforded by holographic light presentations, they compare how compound stimulations differs from individual stimulations, and thus characterise and quantify the phenomena they observe. The study reveals incremental but step-like activations of low threshold Ca2+ spikes, dendritic spikes, and ultimately a more global Na+ spike, as well as identifying which channels are involved by additional pharmacology. Their results also suggest that, unlike larger neurons found elsewhere in the brain, for these neurons, the input locations or the morphological constraints correlate less with the likelihood of generating Na+ and Ca+ spikes. The strength of this study also comes from simultaneous observations of somatic electrophysiological signals and the dendritic Ca2+ signals, as the two types of information can reinforce our understanding of the events that are described.

Overall, the study is important, rigorous, and the data are of high quality. 

The manuscript, however, should be written more accessibly. One reason why this manuscript was extremely difficult to read is the use of many abbreviations. Some abbreviations are common and thus easily understood (names of channels, commonly accepted physiological abbreviations, such as EPSP, Na+ spikes etc). However, others (e.g., sO/I, S1, TPU, SuEPSP/CuEPSP, D-spike, LTS etc etc) are less common. As some of these do not appear frequently in the manuscript, please consider spelling out, and keep abbreviations only to those that are essential. 

Another stylistic aspect that made reading of this manuscript difficult is a slightly convoluted narrative in a few places. For example, when describing the results about response amplitudes that precede a full-blown Na+ spike (page 17), in paragraph 1, the authors describe what "the dendrites know about", only to negate what they write in the last paragraph of the same page. While possibly this could have been the history behind the ultimate interpretations, it is better to state just the latter so as not to confuse readers (unless it is in reference to an earlier study). 

Related to the above points, figures 2-4 mainly describe inputs required for the recruitment of each type of nonlinear events. However, each figure also has a small section on all three types of events. This back-and-forth makes it a bit confusing for readers. In the grand scheme of things (compared to the main points presented in this manuscript), there seems to be too much detail on the observation that, for each type of active events, there is a sudden increase in the Ca2+ signal. 

In the data presented in Fig. 6, observations from the control vs. mibefradil conditions differ between panels d vs. g. What is the difference?

Minor comments

- Fig. 3b: Please provide a brief explanation for the scale bar

- Fig. 4g (left panel): The lines used for the TTX data are hard to see. Please increase the contrast. 

Reviewer #3: In this manuscript, Egger and colleagues use focal, patterned photorelease of glutamate (to activate multiple single spines) of olfactory bulb granule cells to investigate how these cells integrate synaptic information across their dendritic tree. Dendritic integration is a fundamental problem/phenomenon in neuroscience. Olfactory bulb granule cells are particularly intriguing for many reasons - they are inhibitory neurons with dendritic spines, they lack axons, they release transmitter from dendrites (often from reciprocal synapses) and they have local and global processing modes that include nonlinear events mediated by Ca and/or Na. This study covers all these points! The authors demonstrate that activating a small number of spines (synapses) suffices to generate nonlinear events with interesting features that can be tied to information processing in the olfactory bulb circuit. They carefully document the transitions from a linear to (multiple) nonlinear regimes with technically impressive experiments. This work is an important contribution to understanding dendritic integration in olfactory bulb granule cells. 

1. The abstract could be clearer and "punchier". Some suggestions:

lines 20-21: It's not clear to me that placing this idea of inactivation in the abstract is that critical - a general reader may not be expert enough to appreciate it (of course it should be discussed in the Intro/Methods). Perhaps just say "... two-photon uncaging of glutamate at synaptic resolution, along with..." or some such? 

lines 25-26: "until a few spines below global Na+ spike threshold" - this is not clear on initial reading. You mean, of course, until a threshold number of spines are activated… Please clarify.

lines 28-29: The final line seems a bit abrupt and doesn't quite follow easily from the earlier parts. 

2. The authors could consider beginning the introduction with a more general statement about the importance of dendritic integration for information processing throughout the brain (to increase interest for the general audience) and emphasize that olfactory granule cells are a good model system in which to study this since dendritic integration is directly involved in transmitter release.

3. Line 61: "result" should be "results"

4. Line 93, figure 1: What's the reason for the unconventional shapes of the MNI-evoked EPSPs in some cases? Some seem to peak, and then have unusual rising phases (rather than the expected decays over 10s of milliseconds)?

5. line 159: TPU is not defined explicitly here, before or later... at least I couldn't find it!

6. Line 172, figure 2c: I am not fond of the color bars highlighting the categories - why not just color the points themselves? Also in figures further on. [Of course, this is perhaps an esthetic choice]

7. Line 198-199: this result is interesting and perhaps the authors can directly state that there could be Ca regenerative events withOUT Na events?

8. Lines 673-675. The authors write in the methods that "in order to provide optimal optical access to the GC apical dendritic tree, patched GCs were located close to the mitral cell layer (MCL)." It would be helpful to be more specific here. Ideally, if the authors measured the distance from the mitral cell layer for each cell, this should be included. If this was not done, at least an estimate (i.e. <100 μm below the mitral cell layer) would be helpful.

9. Line 492-onwards: Although the authors test other morphological parameters such as spine neck length, we did not see any mention of spine head size. Even if this was not tested as a parameter affecting integration (perhaps it could be tested with data that had already been collected?), it would be helpful to know: how did the authors choose which spines to stimulate? Presumably there was a bias towards larger mushroom spines since filopodial spines would be less easily visualized (although some of the spines in Figure 1a look quite small/filopodial so perhaps this was not the case), but was there a cutoff for the size of the spines chosen? For testing clustered inputs, did the authors choose any spines within the desired distance to stimulate or only large mushroom spines? I realize the authors have already done a lot of regressions, but we are merely curious!

10. Line 582 onwards: The paragraph in the discussion beginning with "All GCs in our sample featured Ca2+-spikes…" (line 582) is confusing to this reader. Please clarify the relationship of this work with your previous work (ref. 5) with regards to local/global calcium spikes. Does glomerular stimulation give different results than uncaging in terms of localization of calcium signals?

11. Line 593: "subthreshold" is misspelled

12. Rats were postnatal day postnatal day 11-21 - were any differences or trends across age observed? Previous work has shown changes in GC properties across this time period (e.g. Dietz and Murthy 2011).

13. Some of the figures are a bit cluttered, confusing and lack legends on the figure. For example, in Figure 2, it would be clearer to put a legend showing that red = AP and green = Ca2+ and then list the spine numbers activated for each trace on the left or right side of the trace rather than having the numbers below the peaks of the traces where they can be lost in the jumble. Ideally magenta should be used instead of red out of consideration for color blindness. In 2a, what does "SLM Vm" mean (should it just be Vm)? Also, what do error bars in this figure mean? Finally, why are different symbols (diamonds, triangles facing to the left or the right or upwards) used in different subplots)? Perhaps simple dots would suffice for all.

14. Please explain Figure 4b, right panel further. Why is the dendrite DF/F below 1 - what is it normalized to? Perhaps it would be clearer to normalize to the average dendrite DF/F for the first for spines as in the left panel of the same figure, unless there is a specific reason to present it differently?

15. A summary panel at the end of Figure 7 showing summation effects for Ca spikes, D-spikes, and global Na spikes could be helpful.

---

## [Decision Letter · Decision Letter 2]

3 Aug 2020

Dear Dr Egger,

Thank you for submitting your revised Research Article entitled "Dendritic integration in olfactory bulb granule cells upon simultaneous multi-spine activation: Low thresholds for non-local spiking activity" for publication in PLOS Biology. I have now obtained advice from the original reviewers and have discussed their comments with the Academic Editor. You will note that reviewer 1, Diego Restrepo, has identified himself. 

We're delighted to let you know that we're now editorially satisfied with your manuscript. However before we can formally accept your paper and consider it "in press", we also need to ensure that your article conforms to our guidelines. A member of our team will be in touch shortly with a set of requests. As we can't proceed until these requirements are met, your swift response will help prevent delays to publication. Please also make sure to address the data and other policy-related requests noted at the end of this email.

*Copyediting*

*Published Peer Review History*

*Early Version*

*Submitting Your Revision*

Sincerely,

Gabriel Gasque, Ph.D.,

Senior Editor,

ggasque@plos.org,

PLOS Biology

DATA POLICY:

-- Please update your Dryad deposition to include data for Fig S2. 

Reviewer remarks:

Reviewer #1, Diego Restrepo: The authors have answered all my questions, and this is an important contribution to the understanding of granule cell function and dendritic integration. I enjoyed reading the section on the potential role of D-spikelets to gamma oscillations. Indeed, regular granule cell APs have long latencies and are unlikely to be involved in regulation of precisely synchronized mitral cell firing. I look forward to see the simulations in the future. 

Reviewer #2: The authors responded to all my comments satisfactorily. Importantly, the manuscript is now well organised - both in terms of the figures as well as the flow. 

Reviewer #3: The authors have addressed all my comments/concerns from the previous review.

---

## [Editor Report · Decision Letter 3]

24 Aug 2020

Dear Dr Egger,

On behalf of my colleagues and the Academic Editor, Justus V Verhagen, I am pleased to inform you that we will be delighted to publish your Research Article in PLOS Biology. 

Early Version

PRESS 

Kind regards,

Alice Musson

Publishing Editor, 

PLOS Biology

on behalf of

Gabriel Gasque,

Senior Editor

PLOS Biology